# Non-stationary Equivariant Graph Neural Networks for Physical Dynamics Simulation

**Chaohao Yuan**[1,2][*], **Maoji Wen**[1][*], **Ercan Engin Kuruoglu**[1][†], **Yang Liu**[2], **Jia Li**[5],
**Tingyang Xu**[3,4], **Deli Zhao**[3], **Hong Cheng**[2], **Yu Rong**[3,4][†]

[1] Tsinghua Shenzhen International Graduate School, Tsinghua University
[2] The Chinese University of Hong Kong
[3] DAMO Academy, Alibaba Group [4] Hupan Lab,
[5] Hong Kong University of Science and Technology (Guangzhou)
chaohaoyuan@link.cuhk.edu.hk, wmj24@mails.tsinghua.edu.cn,
kuruoglu@sz.tsinghua.edu.cn, yu.rong@hotmail.com

## Abstract

To enhance the generalization ability of graph neural networks (GNNs) in learning and simulating physical dynamics, a series of equivariant GNNs have been developed to incorporate the symmetric inductive bias. However, the existing methods do not consider the non-stationarity nature of physical dynamics, where the joint distribution changes over time. Moreover, previous approaches for modeling non-stationary time series typically involve normalizing the data, which disrupts the symmetric assumption inherent in physical dynamics. To model the non-stationary physical dynamics while preserving the symmetric inductive bias, we introduce a **N**on-**S**tationary **E**quivariant **G**raph **N**eural **N**etwork (NS-EGNN) to capture the non-stationarity in physical dynamics while preserving the symmetric property of the model. Specifically, NS-EGNN employs Fourier Transform on segments of physical dynamics to extract time-varying frequency information from the trajectories. It then uses the first and second-order differences to mitigate non-stationarity, followed by pooling for future predictions. Through capturing varying frequency characteristics and alleviate the linear and quadric trend in the raw physical dynamics, NS-EGNN better models the temporal dependencies in the physical dynamics. NS-EGNN has been applied on various types of physical dynamics, including molecular, motion and protein dynamics, and consistently surpasses the existing state-of-the-art algorithms, underscoring its effectiveness. The implementation of NS-EGNN is available at https://github.com/MaojiWEN/NS-EGNN.

## 1 Introduction

Accurately simulating real-world physical dynamics is crucial in numerous fields, including molecular dynamics, motion capture [48], drug discovery [37, 54], and protein folding [1]. The challenge lies in capturing complex interactions among system components. However, traditional methods are either computationally expensive (e.g., Density Functional Theory [21]) or fail to model the complex human intention. Hence, various equivariant Graph Neural Networks (GNNs) [38, 16, 39, 18, 19, 24, 58, 26, 4, 57, 22, 5] have been developed to model such physical interactions while incorporating fundamental symmetry constraints. Specifically, these methods ensure their outputs are equivariant with respect to a specific group, such as E(3), any 3-dimensional translation/orientation/reflection.

---

[*]Equal Contribution.
[†]Corresponding Authors

39th Conference on Neural Information Processing Systems (NeurIPS 2025).

Despite their success, most existing models for physical dynamics focus on single-step frame-to-frame forecasting. That is, they only leverage a single historical frame as input to predict the future states. Such frameworks are insufficient to simulate physical dynamics due to the following issues: (1) **Non-Markovian**. According to the Markovian assumption, future states depend only on the current state and are independent of past states. However, a single frame of the physical dynamics does not comprehensively capture all the details of a given environment because of hidden interactions, such as those involving the solvent; (2) **Non-stationary**. Non-stationarity refers to a dynamic object whose statistical properties and joint distribution change over time. The time-varying distribution in physical dynamics can lead to poor generalization ability in deep learning models. More intuitively, in molecular dynamics, the potential energy of molecules are varying, which results the mean, variance (amplitude of vibration) and covariance (the connections between edges) change over time. Besides, in motion capture dataset, since velocity and physiological state are dynamic, the trajectories of human will also exhibit strong non-stationary property.

It is non-trivial to model Non-stationary and Non-Markovian dynamics. Previous works [28, 30, 52] for modeling non-stationary time-series data mostly adopt normalization approaches to stationarize the series. However, these methods will break the symmetry in the dynamics. An effective alternative, Fourier Transform [41] converts the physical dynamics from time domain to frequency domain, revealing the intensity of each frequency component within the dynamics. The frequency components and power spectrum in the frequency domain can be used to analyze the energy distribution of the physical dynamics [52], which in turn can be used to infer time-domain characteristics, such as variance. As illustrated in Figure 1, the Fourier frequency, $S_1$, $S_2$ and $S_3$, are distinct over different period, exhibiting the dynamic statistical property of dynamics in time domain.

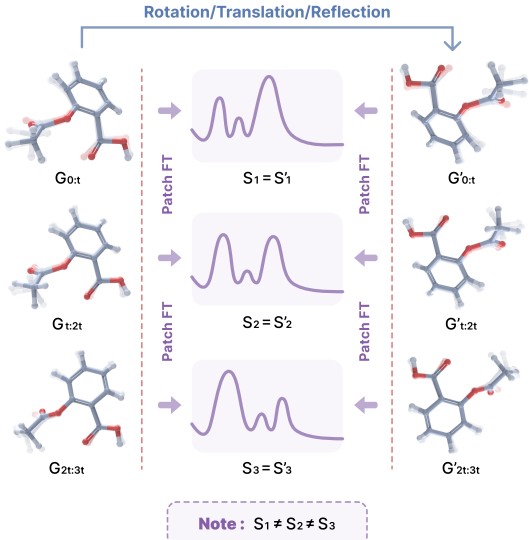

Figure 1: The illustration of non-stationarity and equivariance. **Non-stationarity:** Applying Fourier Transform (FT) to each period of dynamic object will result in different frequency pattern. **Invariance:** With rotation, translation or reflection of physical object, the model should capture the same frequency information.

In this paper, to incorporate the non-stationarity into the equivariant models, we propose **N**on-**S**tationary **E**quivariant **G**raph **N**eural **N**etwork (NS-EGNN), which adopts Fourier transform on patched physical dynamics to capture the varying distribution throughout the entire trajectory. Moreover, the Fourier frequency also reflects the non-Markovian interactions inside the physical systems. Specifically, to capture the dynamic statistical properties inside the non-stationary physical objects, NS-EGNN segments patches of trajectory with overlap and applies Fourier transform to extract the time-varying frequency information from these dynamics. Notably, the extracted frequency features are E(3)-invariant, thus preserving the symmetric properties of the model. Subsequently, to capture the spatial relationships, NS-EGNN employs an equivariant GNN backbone to learn these spatial connections. Moreover, since the Patch Fourier Transform already captures the dynamic patterns, NS-EGNN does not additionally incorporate temporal learning modules, such as attention [47], thus achieving lower computational complexity. Finally, NS-EGNN performs multi-step prediction utilizing the updated spatio-temporal graph through the non-stationary temporal pooling module based on the first order and the second order differencing [32]. Our contributions are summarized as follows:

- We reveal the non-stationary property widely exists in real-world physical objects, such as molecules, proteins and human motions, via Augmented Dickey-Fuller (ADF) [46] test.

- We design a Non-Stationary Equivariant Graph Neural Network (NS-EGNN), leveraging Patch Fourier Transform with window function, to explicitly capture the time-varying frequency information in the dynamic object. Moreover, we design a novel equivariant temporal pooling layer to further alleviate the influence of non-statiaonarity.

- We apply NS-EGNN across various applications, such as molecular-, protein- and macro-level simulation. In the various scenarios, NS-EGNN consistently indicates performance improvements.

## 2 Background

### 2.1 Problem Definition

**Notation.** A physical object can be represented as a graph $\mathcal{G} = (\mathcal{V}, \mathcal{E}, \vec{\mathbf{X}})$. The node features $n_i \in \mathcal{V}$ include non-geometric features $\mathbf{h} \in \mathbb{R}^d$ such as the types of the atoms and a 3D coordinate vector $\vec{\mathbf{X}}_i \in \mathbb{R}^3$, and the edge features $e_{ij} \in \mathcal{E}, \in R^e$ describes the connection between node $i$ and node $j$. In a historical trajectory of this object $\{\mathcal{G}\}_{t=0}^T$, the scalar features $\mathbf{h}$ and edge features $\mathbf{e}$ are constant while the position vectors $\vec{\mathbf{X}}$ change over time.

**Task Definition.** In the trajectory simulation task, given the past trajectory $\{\mathcal{G}\}_{t=0}^T$, the target is to learn a function $f_\theta$ that predicts the future trajectory $\{\mathcal{G}\}_{t=T+1}^{T+\Delta t}$.

$$\{\mathcal{G}\}_{t=T+1}^{T+\Delta t} = f_\theta(\{\mathcal{G}\}_{t=0}^T). \tag{1}$$

Specifically, since only the position vectors are dynamic, the primary focus is on predicting $\vec{\mathbf{X}}$.

### 2.2 Equivariance and Invariance

In the group of $E(3)$, the transformation $g \cdot \vec{\mathbf{X}}$ can be expressed as: $g \cdot \vec{\mathbf{X}} := \mathbf{O}\vec{\mathbf{X}} + \mathbf{t}$, where $\mathbf{O} \in \mathbf{O}(3) = \{\mathbf{O} \in \mathbb{R}^{3 \times 3} | \mathbf{O}^\top \mathbf{O} = \mathbf{I}\}$ represents orthogonal transformations (including rotation and reflection), and $\mathbf{t} \in \mathbb{R}^3$ represents translations.

Given above definitions, to enhance generalization ability, when a physical object undergoes transformations within the group $E(3)$, the equivariant model $f_\theta$ should be able to produce the corresponding prediction for the coordinates. In the context of a spatio-temporal graph, this can be formulated as:

$$\{\mathcal{G} = (\mathcal{V}, \mathcal{E}, g \cdot \vec{\mathbf{X}})\}_{t=T+1}^{T+\Delta t} = f_\theta(\{\mathcal{G} = (\mathcal{V}, \mathcal{E}, g \cdot \vec{\mathbf{X}})\}_{t=0}^T). \tag{2}$$

### 2.3 Non-stationarity

A non-stationary time series exhibits dynamic statistical properties and joint distribution, resulting it difficult to be modeled by deep learning models [51, 36]. Formally, such property can be defined as:

**Definition 1.** *(Non-stationary) Physical dynamics $\{\vec{\mathbf{X}}_t\}$ can be considered as non-stationary if there exists distinct time interval $t_1$ and $t_2$ such that at least one of the following conditions is met: $E(\vec{\mathbf{X}}_{t_1}) \neq E(\vec{\mathbf{X}}_{t_2})$, $Var(\vec{\mathbf{X}}_{t_1}) \neq Var(\vec{\mathbf{X}}_{t_2})$, or $Cov(\vec{\mathbf{X}}_{t_1}, \vec{\mathbf{X}}_{t_1+k}) \neq Cov(\vec{\mathbf{X}}_{t_2}, \vec{\mathbf{X}}_{t_2+k})$ for any lag $k$.*

In other words, if the mean, variance, or covariance function of a physical object evolve over time, then the object is considered to be non-stationary. Furthermore, Fourier frequency also details how the variance of the data is distributed across different frequencies. In the following work, we will utilize this frequency information to model and capture the non-stationary characteristics.

## 3 Methodology

### 3.1 General Framework

As shown in Figure 2, given an EGNN backbone, NS-EGNN consists of Patch Fourier transform (PFT) (Section 3.1.1) and non-stationary pooling layer (NS-Pooling) (Section 3.1.2) to model the non-stationary dynamics equivariantly. Specifically, the brief procedure can be represented as:

$$s = \text{PFT}(\vec{\mathbf{X}}(t)), \tag{3}$$

$$h^{(L)}, s^{(L)}, (\vec{\mathbf{X}}(t)^{(L)})_{t=0}^T = \text{EGNN}(h, s, \vec{\mathbf{X}}(t)_{t=0}^T), \tag{4}$$

$$\vec{\mathbf{X}}^* = \text{NS-Pooling}((\vec{\mathbf{X}}(t)^{(L)})_{t=0}^T). \tag{5}$$

Here, $h^{(L)}, s^{(L)}, (\vec{\mathbf{X}(t)}^{(L)})_{t=0}^{T}$ denote the $L$-th EGNN layer output and $\vec{\mathbf{X}}^*$ is the final pooled trajectory. In PFT, we divide the trajectory into overlapping patches to capture the dynamic variance from the frequency domain, and in NS-Pooling, we employ a difference-based method to minimize the impact by dynamic mean and perform the multi-step prediction by pooling the stationarized dynamics. PFT and NS-Pooling model the dynamics' variance and mean, respectively. First, PFT extracts the invariant frequency features $s$ from the input trajectory. These features, along with the original coordinates $\vec{\mathbf{X}}(t)_{t=0}^{T}$ and scalar features $h$, are fed into the EGNN [39] backbone, as delineated in Section 3.2, which updates the node states. Finally, the NS-Pooling layer is applied to the output coordinates $(\vec{\mathbf{X}}(t)^{(L)})_{t=0}^{T}$ from the EGNN to stationarize the features and make the final multi-step prediction $\vec{\mathbf{X}}^*$. Finally, the training objective of NS-EGNN is the mean square error (MSE) loss $\mathcal{L} = \sum_{t=T}^{T+T_L} \sum_{i=1}^{N} ||\vec{\mathbf{X}}_i^*(t) - \vec{\mathbf{X}}_i(t)||$.

### 3.1.1 Invariant Patch Fourier Transform

**Discrete Fourier Transform (DFT).** DFT is a classic algorithm that converts the trajectory from temporal domain to frequency domain, which contain the periodical information in the physical dynamics. Specifically, DFT $\mathcal{F}$ can extract the frequency information $\vec{\mathbf{s}}_i \in \mathbb{C}^{T \times 3}$ of the physical dynamics at node $i$, and be calculated as follows:

$$\vec{\mathbf{s}}_i(k) = \mathcal{F}(\vec{\mathbf{X}}_i) = \sum_{t=0}^{T} e^{-i' \frac{2\pi}{T} kt} \cdot (\vec{\mathbf{X}}_i(t) - \bar{\vec{\mathbf{X}}}_i(t)),$$
(6)

where $i'$ is the imaginary unit, $k = 0, 1, \cdots, T$ is the frequency index and $\bar{\vec{\mathbf{X}}}_i(t)$ is the average position of all nodes in the graph. Nonetheless, DFT computes a single, global frequency spectrum for the entire trajectory, implicitly assuming the signal's properties are constant over time. We refer to this as *static frequency* information. Due to the inherent non-stationarity of physical dynamics, where statistical properties (and thus frequency components) change over time, this static spectrum is insufficient. Therefore, DFT cannot comprehensively capture the dynamic, time-varying frequency evolution inside the physical objects.

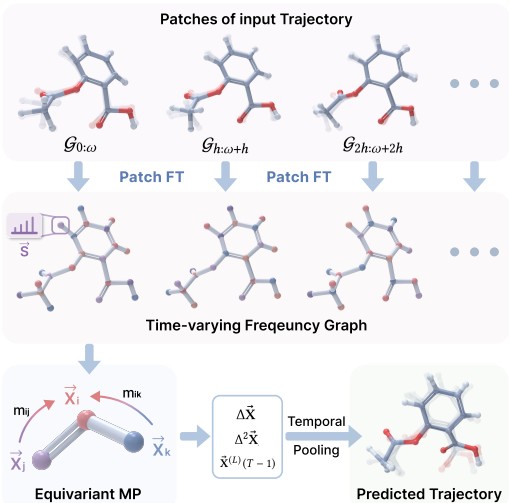

Figure 2: The overall framework of NS-EGNN. NS-EGNN applies window function to the input trajectory and Fourier Transform to receive time-varying frequency graphs. Then, the equivariant GNN will learn from both spatial and spectral information to update the coordinates.

**Patch Fourier Transform (PFT).** To this end, inspired by [15], we design the Patch Fourier Transform (PFT) to extract the frequency content of an object changes over time. Our target is to extract the Fourier frequency at each stage along the trajectory. However, directly cutting off parts of the trajectory can result in spectral leakage [35], which means it might not properly capture the frequency information. Therefore, PFT applies a window function $\mathbf{w}(\cdot; \omega)$ to accurately extract the local frequency in the segmented trajectory, where $\omega \in \mathbb{R}$ denotes window length as a hyperparameter, to determine the number of samples in each segment where Fourier transform is applied, defined as:

$$\text{PFT}(\vec{\mathbf{X}}_i)(p, k) = \sum_{t=0}^{T} e^{-i' \frac{2\pi}{T} kt} \cdot (\vec{\mathbf{X}}_i(t) - \bar{\vec{\mathbf{X}}}_i(t)) \mathbf{w}(t - h \times p; \omega) \tag{7}$$

where $p \in \mathbb{R}$ is the window index of Fourier transform applied, $k \in R$ denotes the frequency index in $p$-th window, and $h \in \mathbb{R}$ denotes the hop size, defined as the number of samples the window is moved forward at each time. Specifically, we adopt the classic Hamming window [35] $\mathbf{w}(t) = 0.54 - 0.46 \cdot \cos(\frac{2\pi t}{\omega - 1})$ as the window function. We also conduct an additional experiment in Appendix C.8 to prove the specific window function will not greatly influence the performance of NS-EGNN. By setting $h < \omega$, the windows overlap, allowing for more thorough capture of the frequency information throughout the trajectory.

**Invariant Frequency Feature.** To capture the entire frequency of physical dynamics, we applied the PFT to all dimensions of coordinates. In each dimension, PFT will result frequency matrix $\mathbf{S}_i \in \mathbb{R}^{r \times K}$, where $r = T/h$ and $K \leq T$ is the number of frequency basis. To integrate the frequency information from different dimensions, the frequency features will be calculated as:

$$\mathbf{s} = \sqrt{\sum_{i=1}^{n} (|\mathbf{S}_i|^2)/n} \tag{8}$$

where $n$ is the number of dimensions. We distribute the frequency feature $\mathbf{s}$ to each node involved in the Fourier transform calculation. Consequently, the dimension of $\mathbf{s}$ is extended to $\mathbb{R}^{T \times K}$, by repeating the feature $h$ times. If a coordinate takes part in multiple Fourier transforms, its corresponding frequency feature will be averaged.

**Multi-Scale PFT.** On the other hand, to prevent the relatively small window size $\omega$ from limiting the ability to capture broader frequencies, we adopt a set of distinct window sizes and hop sizes in the implementation of NS-EGNN. By applying PFT $q$ consecutive times with these varying hyperparameters, NS-EGNN can effectively capture different scales of frequency, enhancing its ability to model non-stationary trajectory. Hence, the final resulted frequency feature is given by $\mathbf{s} \in \mathbb{R}^{q \times T \times K}$.

**Lemma 3.1.** *The extracted frequency feature $\mathbf{s}$ is $E(n)$-invariant.*

The proof can be found in Appendix A.1. In our model, aside from using PFT, we do not incorporate additional temporal modules such as attention mechanisms as PFT already integrates temporal information. An ablation study is set up in Section 4.5. This further enhances efficiency, as the complexity of Fast FT is $O(N \log N)$, while the complexity of attention mechanism is $O(N^2)$, and this is further validated by empirical results in Appendix C.7.

### 3.1.2 Equivariant Non-stationary Temporal Pooling

Inspired by classical statistical algorithms [8], to reduce the non-stationary property, we propose NS-Pooling, to involve differencing the trajectory before pooling:

$$\Delta \vec{\mathbf{X}}_i = [\vec{\mathbf{X}}_i^{(L)}(1) - \vec{\mathbf{X}}_i^{(L)}(0), \vec{\mathbf{X}}_i^{(L)}(2) - \vec{\mathbf{X}}_i^{(L)}(1), \cdots, \vec{\mathbf{X}}_i^{(L)}(T-1) - \vec{\mathbf{X}}_i^{(L)}(T-2)] \tag{9}$$

where $\Delta \vec{\mathbf{X}}_i \in \mathbb{R}^{(T-1) \times 3}$ represents the differentiated. Furthermore, we also derive the second order difference $\Delta^2 \vec{\mathbf{X}}_i = \{\Delta \vec{\mathbf{X}}_i(t) - \Delta \vec{\mathbf{X}}_i(t-1)\}_{t=1}^{T-1}$. Although it is possible to derive higher-order differences, we find that first-order and second-order differences are sufficient for pooling. The corresponding experiments can be found in Appendix C.1. Moreover, while the previous works focused only on predicting the next single frame, our experiments extend the framework to a more challenge setting: forecasting the multi-step trajectory $\vec{\mathbf{X}}_i^* \in \mathbb{R}^{N \times \Delta t \times 3}$ of the physical dynamics, where $\Delta t$ represents the length of the forecasted trajectory. The formulation can be represented as:

$$\vec{\mathbf{X}}_i^* = [\Delta \vec{\mathbf{X}}_i, \Delta^2 \vec{\mathbf{X}}_i] \cdot \gamma + \vec{\mathbf{X}}_i^{(L)}(T-1), \tag{10}$$

where $\gamma \in \mathbb{R}^{(2T-3) \times T_L}$ is a learnable weight matrix. The differencing process removes linear and quadratic trends, making the trajectory more stationary and easier for the model to learn underlying patterns, rather than being distracted by absolute, non-stationary positions.

### 3.1.3 Equivariance Analysis

Let $f_\theta$ denote the overall NS-EGNN models, we have the theorem:

**Theorem 3.2.** *For arbitrary orthogonal transformations and translation vectors $\mathbf{O}, \mathbf{t} \in E(3)$, $f_\theta(\{\mathbf{O}\mathcal{G} + \mathbf{t}\}_{t=0}^T) = \mathbf{O}f_\theta(\{\mathcal{G}\}_{t=0}^T) + \mathbf{t}$.*

The proof is provided in the Appendix A.2. Since the composition of equivariant functions is again equivariant, the PFT module extracts invariant frequency features to feed into the an equivariant backbone. The NS-Pooling layer aggregates the first and second-order differences of the trajectory along with the final position, which is also equivariant.

## 3.2 Spatial Model Backbone

To process the spatial information in the dynamics, we leverage EGNN [39] layers $\mu$ as equivariant backbone in NS-EGNN framework. With the time-varying frequency feature $s$ obtained by PFT, EGNN layers $\mu$ can iteratively update the system states as:

$$\mathbf{h}^{(l+1)}, \mathbf{s}^{(l+1)}, \vec{\mathbf{X}}^{(l+1)} = \mu(\mathbf{h}^{(l)}, \mathbf{s}^{(l)}, \vec{\mathbf{X}}^{(l)}), \tag{11}$$

where $\mathbf{h}_i^{(l)}$, $\mathbf{s}_i^{(l)}$ and $\vec{\mathbf{X}}_i^{(l)}$ are the scalar feature, frequency feature and geometric feature of node $i$ at layer $l$, respectively. Specifically, EGNN employs relative distance $||\vec{\mathbf{X}}_i - \vec{\mathbf{X}}_j||$ as the invariant features:

$$\mathbf{m}_{ij}^{(l)} = f_\theta(\mathbf{h}_i^{(l)}, \mathbf{h}_j^{(l)}, \mathbf{s}_i^{(l)}, \mathbf{s}_j^{(l)}, ||\vec{\mathbf{X}}_i^{(l)} - \vec{\mathbf{X}}_j^{(l)}||^2), \tag{12}$$

where $f_\theta$ is an MLP and $\mathbf{m}_{ij}^{(l)}$ is the invariant message embedding between nodes $i$ and $j$ at $l$-th layer. Given the invariant message embedding, the node coordinates can be updated with equivariance:

$$\vec{\mathbf{X}}_i^{(l+1)} = \vec{\mathbf{X}}^{(l)} + \frac{1}{|\mathcal{N}_i|} \sum_{\substack{j \in \mathcal{N}_i}}^{j \neq i} (\vec{\mathbf{X}}_i^{(l)} - \vec{\mathbf{X}}_j^{(l)}) \phi_x(\mathbf{m}_{ij}^{(l)}), \tag{13}$$

where $\phi_x$ denotes an MLP and $\mathcal{N}_i$ is the neighbors of node $i$ in the physical system. In terms of invariant features, with message embedding $\mathbf{m}$, the hidden representations $\mathbf{h}$ and $\mathbf{s}$ can be updated as:

$$\mathbf{h}_i^{(l+1)} = \mathbf{h}_i^{(l)} + \sum_{\substack{j \in \mathcal{N}_i}}^{j \neq i} \phi_h(\mathbf{h}_j^{(l)}, \mathbf{m}_{ij}^{(l)}), \mathbf{s}_i^{(l+1)} = \mathbf{s}_i^{(l)} + \sum_{\substack{j \in \mathcal{N}_i}}^{j \neq i} \phi_s(\mathbf{s}_j^{(l)}, \mathbf{m}_{ij}^{(l)}), \tag{14}$$

where $f_\theta$, $\phi_h$ and are learnable neural networks.

While we here exploit EGNN [39] as the backbone, the patched dynamics modeling we proposed is a plug-and-play module that can be integrated into other models based on specific scenarios. For instance, in constrained and bounded physical systems, we could develop NS-GMN from GMN [19], and NS-DEGNN from DEGNN [58], respectively. We leave these extensions as future exploration.

## 4 Experiments

### 4.1 Experimental Settings

#### 4.1.1 Datasets & Non-stationary Analysis

We perform experiments on three classic datasets: **1)** MD17 [6], **2)** CMU Motion Capture Database [7], and **3)** AdK equilibrium trajectory dataset [40]. Note that these datasets all exhibit a strong non-stationary property. Specifically, the molecular and protein dynamics in the MD17 and AdK equilibrium trajectory datasets have varying potential energy, which influences the amplitude of vibration. In the motion capture dataset, the human's velocity and physiological state also vary over time.

To confirm the non-stationary nature of the datasets, we conduct Augmented Dickey-Fuller (ADF) tests [10] on each dataset. The ADF test quantifies the degree of stationarity by providing two primary metrics: the *p-value*, which reflects the significance of stationarity, and the *ADF statistic*, where smaller values indicate higher stationarity. Physical dynamics with p-values and ADF statistics below critical thresholds are deemed as stationarity, while higher values suggest the objects possess stronger non-stationary properties. More details can be found in Appendix B.1.

Table 1 summarizes the ADF test results for the MD17, CMU Motion Capture, and AdK

Table 1: Summarized ADF test results for MD17, CMU Motion Capture, and AdK datasets.

| Subset | Non-Stat. Ratio | p-Value Mean | ADF Mean |
|---|---|---|---|
| **MD17 Dataset** | | | |
| Aspirin | 0.8371 | 0.6684 | -0.9185 |
| Benzene | 0.9997 | 0.9112 | 0.0896 |
| Ethanol | 0.9427 | 0.4841 | -1.4999 |
| Malonaldehyde | 0.9972 | 0.7741 | -0.4416 |
| Naphthalene | 0.7710 | 0.3050 | -2.0771 |
| Salicylic | 0.9650 | 0.4540 | -1.6494 |
| Toluene | 0.7840 | 0.2641 | -2.2781 |
| Uracil | 0.4143 | 0.3161 | -2.0884 |
| **CMU Motion Capture Dataset** | | | |
| Walk | 0.9601 | 0.8350 | 1.9866 |
| Run | 0.9449 | 0.7884 | 1.0581 |
| **AdK Dataset** | | | |
| AdK | 0.4873 | 0.1739 | -3.0586 |

Table 2: Averaged prediction error for consecutive forecasts on the MD17 dataset. The reported mean and the standard deviation ($\times 10^{-3}$) are computed over 5 runs.

| | Aspirin | Benzene | Ethanol | Malonaldehyde | Naphthalene | Salicylic | Toluene | Uracil |
|---|---|---|---|---|---|---|---|---|
| ST-TFN | $3.631_{\pm 0.136}$ | $0.823_{\pm 0.007}$ | $1.457_{\pm 0.083}$ | $2.573_{\pm 0.071}$ | $1.171_{\pm 0.061}$ | $2.491_{\pm 0.198}$ | $2.078_{\pm 0.097}$ | $1.753_{\pm 0.037}$ |
| ST-GNN | $10.509_{\pm 2.680}$ | $1.833_{\pm 0.695}$ | $14.349_{\pm 6.393}$ | $4.066_{\pm 0.461}$ | $14.725_{\pm 1.265}$ | $3.064_{\pm 0.212}$ | $2.401_{\pm 0.232}$ | $2.324_{\pm 0.391}$ |
| ST-SE(3)TR | $3.511_{\pm 0.167}$ | $0.848_{\pm 0.035}$ | $1.319_{\pm 0.006}$ | $3.136_{\pm 0.216}$ | $1.063_{\pm 0.006}$ | $2.858_{\pm 0.765}$ | $2.669_{\pm 0.169}$ | $1.754_{\pm 0.038}$ |
| ST-EGNN | $3.257_{\pm 0.394}$ | $0.876_{\pm 0.144}$ | $0.879_{\pm 0.112}$ | $1.878_{\pm 0.258}$ | $0.922_{\pm 0.063}$ | $1.909_{\pm 0.320}$ | $1.491_{\pm 0.139}$ | $1.545_{\pm 0.152}$ |
| EqMotion | $3.790_{\pm 0.018}$ | $1.166_{\pm 0.279}$ | $1.882_{\pm 0.011}$ | $2.793_{\pm 0.013}$ | $3.201_{\pm 0.008}$ | $3.258_{\pm 0.004}$ | $2.917_{\pm 0.056}$ | $3.288_{\pm 0.002}$ |
| STGCN | $4.175_{\pm 0.171}$ | $1.001_{\pm 0.063}$ | $214.904_{\pm 0.076}$ | $3.455_{\pm 0.370}$ | $3.454_{\pm 0.104}$ | $3.433_{\pm 0.052}$ | $3.110_{\pm 0.131}$ | $3.576_{\pm 0.112}$ |
| AGL-STAN | $587.048_{\pm 73.836}$ | $5.914_{\pm 2.247}$ | $303.185_{\pm 83.200}$ | $53.283_{\pm 23.115}$ | $33.055_{\pm 7.606}$ | $3.256_{\pm 0.310}$ | $8.338_{\pm 1.475}$ | $10.509_{\pm 0.351}$ |
| ESTAG | $0.740_{\pm 0.059}$ | $0.072_{\pm 0.014}$ | $0.475_{\pm 0.020}$ | $0.874_{\pm 0.179}$ | $0.405_{\pm 0.020}$ | $0.636_{\pm 0.100}$ | $0.376_{\pm 0.043}$ | $0.533_{\pm 0.033}$ |
| NS-EGNN | $\mathbf{0.421_{\pm 0.023}}$ | $\mathbf{0.050_{\pm 0.008}}$ | $\mathbf{0.410_{\pm 0.010}}$ | $\mathbf{0.589_{\pm 0.035}}$ | $\mathbf{0.275_{\pm 0.023}}$ | $\mathbf{0.387_{\pm 0.093}}$ | $\mathbf{0.308_{\pm 0.039}}$ | $\mathbf{0.379_{\pm 0.027}}$ |

Table 3: Final prediction error for consecutive forecasts on the MD17 dataset. Bold font indicates the best result. The reported mean and the standard deviation ($\times 10^{-3}$) are computed over 5 runs.

| | Aspirin | Benzene | Ethanol | Malonaldehyde | Naphthalene | Salicylic | Toluene | Uracil |
|---|---|---|---|---|---|---|---|---|
| ST-TFN | $6.026_{\pm 0.505}$ | $1.615_{\pm 0.009}$ | $2.051_{\pm 0.269}$ | $4.596_{\pm 0.307}$ | $1.436_{\pm 0.061}$ | $3.571_{\pm 0.273}$ | $2.700_{\pm 0.365}$ | $2.893_{\pm 0.089}$ |
| ST-GNN | $20.818_{\pm 7.168}$ | $3.059_{\pm 0.513}$ | $16.586_{\pm 6.768}$ | $8.084_{\pm 1.017}$ | $8.361_{\pm 5.020}$ | $4.276_{\pm 1.172}$ | $3.420_{\pm 0.140}$ | $3.054_{\pm 0.329}$ |
| ST-SE(3)TR | $7.177_{\pm 1.037}$ | $1.941_{\pm 0.672}$ | $1.700_{\pm 0.029}$ | $6.769_{\pm 0.599}$ | $1.463_{\pm 0.049}$ | $3.428_{\pm 0.629}$ | $2.868_{\pm 0.183}$ | $2.870_{\pm 0.083}$ |
| ST-EGNN | $4.387_{\pm 0.635}$ | $1.162_{\pm 0.238}$ | $1.161_{\pm 0.100}$ | $2.172_{\pm 0.678}$ | $1.097_{\pm 0.126}$ | $2.559_{\pm 0.307}$ | $1.673_{\pm 0.315}$ | $2.127_{\pm 0.328}$ |
| EqMotion | $7.665_{\pm 0.056}$ | $2.153_{\pm 0.064}$ | $2.975_{\pm 0.011}$ | $5.489_{\pm 0.005}$ | $5.695_{\pm 0.036}$ | $6.248_{\pm 0.012}$ | $4.898_{\pm 0.011}$ | $6.423_{\pm 0.122}$ |
| STGCN | $7.628_{\pm 0.037}$ | $2.008_{\pm 0.037}$ | $2.957_{\pm 0.015}$ | $5.516_{\pm 0.047}$ | $5.659_{\pm 0.038}$ | $6.253_{\pm 0.040}$ | $4.880_{\pm 0.006}$ | $6.354_{\pm 0.004}$ |
| AGL-STAN | $634.663_{\pm 99.469}$ | $6.028_{\pm 2.081}$ | $271.624_{\pm 66.230}$ | $58.795_{\pm 7.170}$ | $29.899_{\pm 10.410}$ | $3.798_{\pm 0.491}$ | $8.418_{\pm 2.263}$ | $10.085_{\pm 0.350}$ |
| ESTAG | $1.446_{\pm 0.225}$ | $0.169_{\pm 0.058}$ | $0.898_{\pm 0.068}$ | $1.442_{\pm 0.075}$ | $0.801_{\pm 0.074}$ | $1.038_{\pm 0.098}$ | $0.730_{\pm 0.111}$ | $1.086_{\pm 0.150}$ |
| NS-EGNN | $\mathbf{0.833_{\pm 0.187}}$ | $\mathbf{0.147_{\pm 0.027}}$ | $\mathbf{0.695_{\pm 0.020}}$ | $\mathbf{0.988_{\pm 0.055}}$ | $\mathbf{0.598_{\pm 0.149}}$ | $\mathbf{0.518_{\pm 0.026}}$ | $\mathbf{0.380_{\pm 0.056}}$ | $\mathbf{0.719_{\pm 0.219}}$ |

datasets. For the MD17 dataset, molecules such as Benzene exhibit higher p-values and positive ADF statistics, reflecting pronounced non-stationary behavior. The CMU Motion Capture dataset's walk and run subsets show high p-values and positive ADF statistics, confirming their non-stationary nature. Lastly, the AdK dataset demonstrates substantial variability in its temporal dynamics, with approximately half of its nodes classified as non-stationary.

### 4.1.2 Baselines

We compare the performance of our proposed model with several widely used baselines in spatio-temporal trajectory modeling. **STGCN** [53] adopts a spatio-temporal convolutional architecture and is adjusted to predict residual coordinates between frames rather than absolute positions, as directly predicting the latter often leads to suboptimal performance. AGL-STAN [42], which combines adaptive graph learning with self-attention mechanisms, is modified to handle weighted temporal aggregation to better capture intra-temporal dependencies. ST-GNN [14], **ST-SE(3)-Transformer** [12], denoted as **ST-SE(3)TR**, **ST-TFN** [43], and **ST-EGNN** [39] are included as GNN baselines. Except for ST-GNN, which is based solely on the message passing framework, other approaches leverage rotational and translational invariance for trajectory prediction. Another representative model, **EqMotion** [48], integrates spatio-temporal information using attention-based fusion for modeling the object dynamics. **ESTAG** [47] first models the non-Markovian nature of physical dynamics and proposes an equivariant temporal attention module to capture the latent interaction. All baselines are modified to process the full historical trajectory (e.g., using linear encoders or, in the case of ESTAG, its native temporal attention) to ensure a fair comparison against our multi-step input model.

### 4.2 Molecular Dynamics

**Setting.** We evaluate the performance of our proposed model on the MD17 dataset, which includes molecular trajectories generated by MD simulation. The length of the input time series is set to 100, predicting the next 10 timesteps, with $\Delta t = 5$, as time series requires more timesteps to observe non-stationarity. We also conduct the experiments with fewer timesteps in Appendix C.5, aligning with the setting in ESTAG, which also has satisfactory performance improvement. The dataset is split into training, validation, and testing sets with ratios of 0.2, 0.4, and 0.4, respectively.

**Evaluation Metrics.** We use two standard evaluation metrics: 1) Average Displacement Error (ADE), which measures the average $\ell_2$ distance between the predicted and ground truth molecular trajectories over all timesteps. 2) Final Displacement Error (FDE), which evaluates the $\ell_2$ distance between the predicted and ground truth positions at the final predicted step.

**Results.** Tables 2 and 3 summarize the ADE and FDE across all models. Notably, NS-EGNN emerges as the most effective model, surpassing baseline models considerably in both ADE and FDE metrics.

Table 4: Prediction error (MSE) for Walk ($\times 10^{-1}$) and Run ($\times 10^{0}$) cases under different time intervals (5ts, 10ts, 15ts, 20ts). The reported mean and standard deviation are computed over 5 runs.

| Dataset | Walk ($\times 10^{-1}$) | | | | | Run ($\times 10^{0}$) | | | | |
|---|---|---|---|---|---|---|---|---|---|---|
| Time Itv. | 5ts | 10ts | 15ts | 20ts | Average | 5ts | 10ts | 15ts | 20ts | Average |
| ST-GNN | $1.121_{\pm 0.159}$ | $1.224_{\pm 0.100}$ | $2.615_{\pm 0.478}$ | $3.359_{\pm 0.601}$ | $1.941_{\pm 0.335}$ | $0.560_{\pm 0.107}$ | $1.160_{\pm 0.166}$ | $1.538_{\pm 0.234}$ | $1.779_{\pm 0.278}$ | $1.259_{\pm 0.196}$ |
| ST-TFN | $0.238_{\pm 0.032}$ | $0.721_{\pm 0.038}$ | $1.320_{\pm 0.067}$ | $2.092_{\pm 0.094}$ | $1.093_{\pm 0.058}$ | $0.396_{\pm 0.073}$ | $0.796_{\pm 0.054}$ | $1.708_{\pm 0.318}$ | $2.086_{\pm 0.133}$ | $1.247_{\pm 0.145}$ |
| ST-SE(3)TR | $0.146_{\pm 0.017}$ | $0.376_{\pm 0.097}$ | $0.760_{\pm 0.161}$ | $1.119_{\pm 0.347}$ | $0.600_{\pm 0.156}$ | $0.280_{\pm 0.045}$ | $0.700_{\pm 0.131}$ | $1.165_{\pm 0.267}$ | $1.732_{\pm 0.550}$ | $0.969_{\pm 0.248}$ |
| ST-EGNN | $0.188_{\pm 0.026}$ | $0.591_{\pm 0.103}$ | $1.140_{\pm 0.123}$ | $2.097_{\pm 0.205}$ | $0.979_{\pm 0.114}$ | $0.444_{\pm 0.072}$ | $1.082_{\pm 0.113}$ | $2.375_{\pm 0.202}$ | $3.784_{\pm 0.429}$ | $1.921_{\pm 0.204}$ |
| EqMotion | | | – | | | $21.074_{\pm 2.073}$ | $15.299_{\pm 4.127}$ | $21.074_{\pm 2.073}$ | $18.604_{\pm 3.503}$ | $19.013_{\pm 2.944}$ |
| STGCN | $0.302_{\pm 0.115}$ | $0.828_{\pm 0.203}$ | $1.516_{\pm 0.384}$ | $1.988_{\pm 0.206}$ | $1.159_{\pm 0.228}$ | $0.131_{\pm 0.024}$ | $0.582_{\pm 0.121}$ | $1.101_{\pm 0.089}$ | $1.508_{\pm 0.176}$ | $0.831_{\pm 0.103}$ |
| AGL-STAN | $1.729_{\pm 0.516}$ | $1.789_{\pm 0.673}$ | $2.030_{\pm 0.704}$ | $2.155_{\pm 0.763}$ | $1.926_{\pm 0.664}$ | $0.511_{\pm 0.137}$ | $0.628_{\pm 0.191}$ | $\underline{0.648}_{\pm 0.271}$ | $\mathbf{0.831}_{\pm 0.283}$ | $\underline{0.654}_{\pm 0.221}$ |
| ESTAG | $\underline{0.054}_{\pm 0.004}$ | $\underline{0.213}_{\pm 0.012}$ | $\underline{0.530}_{\pm 0.038}$ | $\underline{1.085}_{\pm 0.070}$ | $\underline{0.471}_{\pm 0.031}$ | $\underline{0.041}_{\pm 0.002}$ | $\underline{0.250}_{\pm 0.019}$ | $0.771_{\pm 0.050}$ | $1.767_{\pm 0.251}$ | $0.707_{\pm 0.081}$ |
| **NS-EGNN** | $0.051_{\pm 0.002}$ | $0.166_{\pm 0.006}$ | $0.397_{\pm 0.037}$ | $0.775_{\pm 0.085}$ | $0.347_{\pm 0.033}$ | $0.033_{\pm 0.002}$ | $0.187_{\pm 0.009}$ | $0.584_{\pm 0.076}$ | $\underline{1.226}_{\pm 0.215}$ | $0.508_{\pm 0.076}$ |

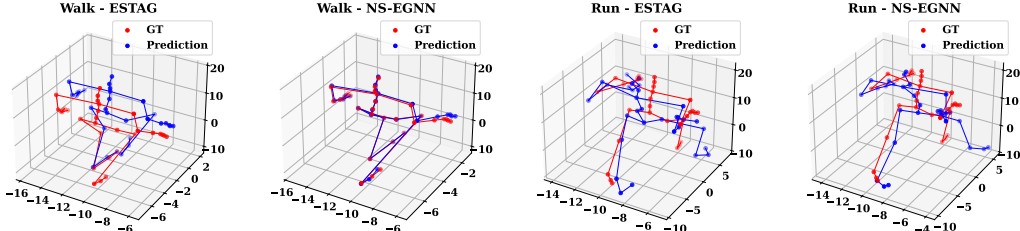

Figure 3: Visualization of predicted trajectories for *run* and *walk* motions at *time gap* = 10 and 20. Ground truth is represented in red, and predictions are in blue.

Specifically, compared with the current state-of-the-art model, NS-EGNN achieves an impressive relative improvement of **29.77**% on ADE and **33.33**% on FDE. Particularly with complex molecules like Aspirin and Malonaldehyde, NS-EGNN delivers an even more pronounced enhancement, reaching a **42.74**% relative improvement for Aspirin. The second-best algorithm, which also utilizes a spatio-temporal approach to capture physical dynamics, underscores the importance of leveraging historical trajectory data. Nevertheless, NS-EGNN excels by explicitly modeling the dynamic statistical properties of physical objects, thereby achieving the best performance. Baseline models like STGCN and AGL-STAN exhibit high errors, as they are not equivariant. AGL-STAN's particularly poor performance, consistent with findings in other works [47], is due to its non-equivariant architecture struggling to model symmetric physical dynamics. Equivariant baselines such as ST-SE(3)TR and ST-EGNN perform better but are less robust in capturing the dynamic behaviors of larger molecules.

## 4.3 Motion Capture

**Setting.** CMU Motion Capture Database involves the trajectories of human motion under various scenarios. For this experiment, we focus on the *walk* and *run* motions, selecting trajectories with sufficient length. The input sequence consists of the past 10 frames used to predict the subsequent frame, with a data frame interval of $\Delta t = 1$. Additionally, we introduce a hyperparameter *time gap*, representing the delay between the last observed frame and the target frame to be predicted. Experiments are conducted with four values of *time gap*: 5ts, 10ts, 15ts, and 20ts. In *run* dataset, we additionally normalize the dataset, following previous works [47].

**Results.** Table 4 presents MSE across different models for both *walk* and *run* motions, indicating NS-EGNN consistently achieves the best performance in most settings. Specifically, NS-EGNN achieves an average improvement of **21.52%** in *walk* and **15.29%** in *run*, leading to an overall relative gain of **18.41%** across the entire dataset. This highlights the model's ability to capture the temporal and spatial dependencies crucial for human motion prediction. While AGL-STAN achieves the best performance in one case, NS-EGNN also remains highly competitive in that case. Overall, NS-EGNN demonstrates strong performance in both short-term and long-term predictions across different motions. Given that the motion dataset exhibits strong non-stationarity, it is not surprising that NS-EGNN achieves superior performance.

**Visualization.** Additionally, we provide visualizations of the predicted trajectories compared to the ground truth for both motions at *time gap* equals 10ts and 20ts, respectively. The visualizations, shown in Figure 3, include predictions from both ESTAG and our NS-EGNN. As illustrated, NS-EGNN predictions align more closely with the ground truth trajectories compared to ESTAG, particularly for complex joint movements. In the *walk* scenario, NS-EGNN exhibits smoother and more stable predictions, with trajectory paths closely following the ground truth points. For the *run*

motion, ESTAG struggles with maintaining coherence in the limb trajectories, resulting in noticeable distortions and erratic movements. In contrast, NS-EGNN better preserves the structural integrity of the motion, particularly in the torso and overall body posture, leading to a more realistic and physically plausible trajectory.

Table 5: Mean Squared Error (MSE) on the AdK dataset for protein dynamics prediction. The reported mean MSE values are computed over 5 runs. Bold font indicates the best result.

| Model | STGCN | ST-GNN | ST-EGNN | ST-GMN | AGL-STAN | ESTAG | **NS-EGNN** |
|---|---|---|---|---|---|---|---|
| **MSE** | 3.007 | 2.267 | 1.751 | 1.743 | 1.853 | 1.758 | **1.738** |

## 4.4 Protein Dynamics

**Setting.** We evaluate our model on the AdK protein dynamics dataset, which involves predicting protein motions. For this dataset, we disable NS-pooling because it tends to overfit to noise. The input sequence consists of 60 timesteps used to predict the subsequent 10 timesteps, with $\Delta t = 5$. The dataset is split into training, validation, and testing sets with ratios of 0.6, 0.2, and 0.2, respectively. For consistency, all models are configured with 4 layers.

**Results.** Table 5 presents the MSE of all models on the AdK dataset. While the overall improvement margin is less pronounced than in previous experiments, NS-EGNN still achieves the best performance with an MSE of 1.738. This aligns with our findings from the ADF test (Table 1), where the AdK dataset exhibits a smaller proportion of non-stationary nodes compared to other datasets. As a result, the impact of explicitly addressing non-stationarity is naturally less significant. This further demonstrates NS-EGNN's generalization ability, allowing it to make accurate predictions even in datasets where non-stationary effects are less dominant.

Table 6: Ablation study results on the MD17 dataset. The table reports ADE values ($\times 10^{-3}$), averaged over three runs. Lower values indicate better performance.

| Method | Aspirin | Benzene | Ethanol | Malonaldehyde | Naphthalene | Salicylic | Toluene | Uracil | Average |
|---|---|---|---|---|---|---|---|---|---|
| **NS-EGNN** | **0.421** | 0.050 | 0.407 | **0.600** | **0.270** | **0.387** | **0.308** | 0.379 | **0.353** |
| w/ Attention | 0.499 | **0.039** | **0.389** | 0.751 | 0.280 | 0.446 | 0.317 | **0.377** | 0.387 |
| w/o Patched FT | 0.948 | 0.076 | 0.547 | 0.876 | 0.360 | 0.499 | 0.431 | 0.504 | 0.530 |
| w/o Differentiation | 2.622 | 0.265 | 0.700 | 1.929 | 0.949 | 2.249 | 1.103 | 1.448 | 1.408 |
| w/o Equivariance | 225.608 | 655.109 | 220.134 | 871.211 | 1365.762 | 10.397 | 281.836 | 7.861 | 454.740 |

## 4.5 The Effectiveness of Each Component in NS-EGNN

To evaluate the contributions of key components in NS-EGNN, we conduct experiments including or removing specific modules to analyze their impact, and results are in Table 6. From the table, we derive the following conclusions: **1) PFT extracts expressive temporal features.** Prior work [47] introduced an equivariant temporal attention mechanism for learning physical dynamics. To explore potential improvements, we also incorporate this module. However, the overall performance shows a slight decline, suggesting that PFT already generates sufficiently expressive temporal features. **2) PFT is capable of capturing the dynamic statistic property in the non-stationary trajectory.** We replace the PFT with a standard Fourier Transform applied to the entire trajectory. The observed worse performance suggests that PFT plays a critical role in modeling the temporal variance of non-stationary trajectories. This highlights the necessity of preserving localized frequency patterns. **3) Differentiation greatly reduces the negative impact of linear and quadric trend in the raw physical dynamics.** The removal of this module results in a noticeable decline in performance, underscoring the critical role of differentiation in addressing mean shifts within non-stationary dynamics. Furthermore, without this component, the model loses crucial equivariant property, leading to suboptimal predictions. **4) Equivariant backbone enhances the generalization ability in dynamic simulation.** The standard message passing GNN struggles to capture the symmetrical properties inherent in Euclidean space, leading to significantly poor performance. We further provide the NS-EGNN removing Fourier Transform entirely in Appendix C.4.

# 5 Related Works

**Graph Neural Networks for Geometric Trajectory.** TFN [43] and SE(3)-Transformer [12] employ spherical harmonics to extract high-order geometric representations. To reduce the computation in high-order representations, EGNN [39] updates invariant messages using relative distances and then derives directional vectors from these messages. For specific scenarios, many variants of EGNN [27] have been proposed, such as SGNN [17], GMN [19], DEGNN [58], EGHN [18] and EGNO [49]. EqMotion [48] and ESTAG [47] propose extracting the invariant and equivariant geometric features from historical dynamics, capturing temporal dependencies. However, these methods overlook the non-stationarity of physical dynamics, leading to suboptimal performance.

**Non-stationarity in Time Series Forecasting.** Before deep learning models, the classic ARIMA [3, 2] algorithm addressed non-stationarity by differencing the time series. The varying distribution of non-stationary data presents additional challenges for deep models. Pre-processing offers a straightforward and effective method to make the time series stationary. Adaptive Norm [33] normalizes fragment of the series according to the global statistics. Moreover, DAIN [34], RevIN [20], SAN [29], FAN [52], DDN [9] and IN-Flow [11] introduce learnable neural networks to normalize time series data. Furthermore, Transformers [28, 23, 13, 45, 44, 56, 55, 25] have been utilized to model non-stationary series with specialized attention mechanisms. However, direct stationarization of physical object coordinates of physical objects will unavoidably break the symmetric property of the deep models. To bridge this gap, NS-EGNN captures dynamic distributions while maintaining equivariance.

# 6 Conclusion

We show that trajectories of physical dynamics are highly non-stationary. Our NS-EGNN framework leverages the PFT and NS-Pooling modules to capture these patterns. By modeling non-stationary physical dynamics with equivariance, NS-EGNN greatly outperform the SOTA methods. A limitation is diminished gains on stationary data, where repeated FT increases computation with marginal benefits. Future work could extend NS-EGNN to domains to various settings and explore more spatial backbones [50] according to the specific scenarios. Furthermore, other methods for modeling non-stationary data, such as wavelet transforms, also break the equivariance. However, the adaption of these advanced method into equivariance also worth exploration.

## Acknowledgments

This work is supported by Shenzhen Science and Technology Innovation Commission under Grant JCYJ20220530143002005, Tsinghua Shenzhen International Graduate School Start-up fund under Grant QD2022024C, Shenzhen Ubiquitous Data Enabling Key Lab under Grant ZDSYS20220527171406015, Damo Academy (Hupan Laboratory) through Damo Academy (Hupan Laboratory) Innovative Research Program, Damo Academy through Damo Academy Research Intern Program, Research Grants Council of the Hong Kong Special Administrative Region, China (No. CUHK 14217622), and Guangzhou Industrial Information and Intelligent Key Laboratory Project (No. 2024A03J0628). Additionally, Chaohao Yuan would like to thank his fiancée, Siying Xu, for her companionship and great help in visualization (Figure 1 & 2) in this work.

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

# A Proofs

## A.1 Proof of Lemma 3.1

**Lemma 3.1** (,). *The extracted frequency feature* **s** *is* $E(n)$*-invariant.*

*Proof.* Recall Eq. 7,

$$\text{PFT}(\vec{\mathbf{X}}_i)(p, k) = \sum_{t=0}^{T} e^{-i' \frac{2\pi}{T} kt} \cdot (\vec{\mathbf{X}}_i(t) - \bar{\vec{\mathbf{X}}}_i(t)) \mathbf{w}(t - h \times p; \omega) \tag{15}$$

After E(3) transformation, orthogonal **O** combined with translation **t** to the trajectory, the updated Fourier frequency is denoted as $\mathbf{S}_i^e$:

$$\mathbf{S}_i^e = \sum_{t=0}^{T} e^{-i' \frac{2\pi}{T} kt} \cdot (\mathbf{O}\vec{\mathbf{X}}_i(t) + \mathbf{t} - \mathbf{O}\bar{\vec{\mathbf{X}}}_i(t) - \mathbf{t}) \mathbf{w}(t - h \times p; \omega) \tag{16}$$

$$= \sum_{t=0}^{T} e^{-i' \frac{2\pi}{T} kt} \cdot \mathbf{O}(\vec{\mathbf{X}}_i(t) - \bar{\vec{\mathbf{X}}}_i(t)) \mathbf{w}(t - h \times p; \omega) \tag{17}$$

$$= \mathbf{O} \sum_{t=0}^{T} e^{-i' \frac{2\pi}{T} kt} \cdot (\vec{\mathbf{X}}_i(t) - \bar{\vec{\mathbf{X}}}_i(t)) \mathbf{w}(t - h \times p; \omega) \tag{18}$$

$$= \mathbf{O}\mathbf{S}_i \tag{19}$$

We denote the transformed frequency feature as $\mathbf{s}^e$ with $\mathbf{O}\mathbf{O}^T = \mathbf{I}$,

$$\mathbf{s}^e = \sqrt{\sum_{i=1}^{n} (|\mathbf{S}_i^e|^2)/n} \tag{20}$$

$$= \sqrt{\sum_{i=1}^{n} (|\mathbf{O}\mathbf{S}_i|^2)/n} \tag{21}$$

$$= \sqrt{\sum_{i=1}^{n} (|\mathbf{S}_i|^2)/n} \tag{22}$$

$$= \mathbf{s}. \tag{23}$$

$\square$

## A.2 Proof of Theorem 3.2

**Lemma A.1.** *Equivariant Message Passing is* $E(n)$*-equivariant.*

*Proof.*

$$\mathbf{m}_{ij}^{(l)} = f_\theta(\mathbf{h}_i^{(l)}, \mathbf{h}_j^{(l)}, \mathbf{s}_i^{(l)}, \mathbf{s}_j^{(l)}, ||(\mathbf{O}\vec{\mathbf{X}}_i^{(l)} + \mathbf{t}) - (\mathbf{O}\vec{\mathbf{X}}_j^{(l)} + \mathbf{t})||^2) \tag{24}$$

$$= f_\theta(\mathbf{h}_i^{(l)}, \mathbf{h}_j^{(l)}, \mathbf{s}_i^{(l)}, \mathbf{s}_j^{(l)}, ||\mathbf{O}(\vec{\mathbf{X}}_i^{(l)} - \vec{\mathbf{X}}_j^{(l)})||^2) \tag{25}$$

$$= f_\theta(\mathbf{h}_i^{(l)}, \mathbf{h}_j^{(l)}, \mathbf{s}_i^{(l)}, \mathbf{s}_j^{(l)}, ||(\vec{\mathbf{X}}_i^{(l)} - \vec{\mathbf{X}}_j)^{(l)}||^2) \tag{26}$$

$$\mathbf{O}\vec{\mathbf{X}}_i^{(l+1)} + \mathbf{t} = \mathbf{O}\vec{\mathbf{X}}^{(l)} + \mathbf{t} + \frac{1}{|\mathcal{N}_i|} \sum_{j \in \mathcal{N}_i}^{j \neq i} (\mathbf{O}\vec{\mathbf{X}}_i^{(l)} + \mathbf{t} - \mathbf{O}\vec{\mathbf{X}}_j^{(l)} - \mathbf{t}) \phi_x(\mathbf{m}_{ij}^{(l)}). \tag{27}$$

$$= \mathbf{O}(\vec{\mathbf{X}}^{(l)} + \frac{1}{|\mathcal{N}_i|} \sum_{\substack{j \in \mathcal{N}_i \\ j \neq i}} (\vec{\mathbf{X}}_i^{(l)} - \vec{\mathbf{X}}_j^{(l)}) \phi_x(\mathbf{m}_{ij}^{(l)})) + \mathbf{t} \tag{28}$$

$\square$

**Lemma A.2.** *Equivariant Temporal Pooling is E(n)-equivariant.*

*Proof.*

$$\vec{\mathbf{X}}_i^* = [\Delta\vec{\mathbf{X}}_i, \Delta^2\vec{\mathbf{X}}_i] \cdot \gamma + \vec{\mathbf{X}}_i^{(L)}(T-1). \tag{29}$$

With orthogonal transformation $\mathbf{O}$ and translation $\mathbf{t}$ in E(n) group.

$$\mathbf{O}\Delta\vec{\mathbf{X}}_i = \{\mathbf{O}\vec{\mathbf{X}}_i(t) + \mathbf{t} - \mathbf{O}\vec{\mathbf{X}}_i(t-1) - \mathbf{t}\}_{t=1}^{T} \tag{30}$$
$$= \mathbf{O}\{\vec{\mathbf{X}}_i(t) - \vec{\mathbf{X}}_i(t-1)\}_{t=1}^{T} \tag{31}$$

$$\mathbf{O}\Delta^2\vec{\mathbf{X}}_i = \mathbf{O}\{\Delta\vec{\mathbf{X}}_i(t) - \Delta\vec{\mathbf{X}}_i(t-1)\}_{t=0}^{T-1} \tag{32}$$
$$= \{\mathbf{O}\Delta\vec{\mathbf{X}}_i(t) - \mathbf{O}\Delta\vec{\mathbf{X}}_i(t-1)\}_{t=0}^{T-1} \tag{33}$$

Then, we have

$$\mathbf{O}\vec{\mathbf{X}}_i^* + \mathbf{t} = [\mathbf{O}\Delta\vec{\mathbf{X}}_i, \mathbf{O}\Delta^2\vec{\mathbf{X}}_i] \cdot \gamma + \mathbf{O}\vec{\mathbf{X}}_i^{(L)}(T-1) + \mathbf{t} \tag{34}$$

$\square$

**Theorem 3.2** (,). *For arbitrary orthogonal transformations and translation vectors* $\mathbf{O}, \mathbf{t} \in E(3)$, $f_\theta(\{\mathbf{O}\mathcal{G} + \mathbf{t}\}_{t=0}^T) = \mathbf{O}f_\theta(\{\mathcal{G}\}_{t=0}^T) + \mathbf{t}$.

*Proof.* As shown in Lemma 3.1, Lemma A.1, and Lemma A.2, since extracted spectral feature is invariant, and the rest of two components, Equivariant Message Passing and Equivariant Temporal Pooling, are equivariant, the model, $f_\theta(\cdot)$, is equivariant as well. $\square$

# B Implementation Details

## B.1 More Details on ADF Test

To provide a more comprehensive view of the Augmented Dickey-Fuller (ADF) test results, we present additional statistical measures that are omitted in the main paper for brevity.

To mitigate the influence of outliers, we exclude the top and bottom 3% of values when computing the mean and standard deviation of the ADF test metrics. For each node's spatial coordinates $x, y, z$, we determine the stationarity individually. A node is classified as non-stationary if any of its $x$, $y$, or $z$ dimensions fails the stationarity test. This ensures a comprehensive evaluation of the node's temporal behavior across all spatial dimensions.

Table 8 provides detailed statistics for each dataset, including the total number of trajectories, the count of non-stationary trajectories, the mean and standard deviation of the p-value, as well as the mean and standard deviation of the ADF statistic.

Table 7: Critical values for the ADF test across different datasets. Each column represents the 1%, 5%, and 10% significance thresholds.

| Dataset | Mean 1% | Mean 5% | Mean 10% |
|---------|---------|---------|----------|
| MD17 | -3.444 | -2.868 | -2.570 |
| Motion Walk | -5.0538 | -3.5246 | -2.858 |
| Motion Run | -5.0896 | -3.5391 | -2.8633 |
| AdK | -3.5521 | -2.9144 | -2.5949 |

Table 7 lists the critical values used in the ADF test across different significance levels (1%, 5%, and 10%) for various datasets. If the ADF statistic of a time series is lower than the critical value at a given significance level, the null hypothesis of non-stationarity is rejected, confirming stationarity.

Table 8: Detailed ADF test results for the MD17, CMU Motion Capture, and AdK datasets. Each row summarizes key statistical measures for different subsets.

| Subset | Total Trajectories | Non-Stationary Trajectories | Mean p-value | Std p-value | Mean ADF Statistic | Std ADF Statistic |
|---|---|---|---|---|---|---|
| **MD17 Dataset** | | | | | | |
| Aspirin | 6500 | 5441 | 0.6684 | 0.2641 | -0.9185 | 1.0372 |
| Benzene | 3000 | 2999 | 0.9112 | 0.1233 | 0.0896 | 0.8995 |
| Ethanol | 1500 | 1414 | 0.4841 | 0.3082 | -1.4999 | 1.0187 |
| Malonaldehyde | 2500 | 2493 | 0.7741 | 0.2464 | -0.4416 | 1.1369 |
| Naphthalene | 5000 | 3855 | 0.305 | 0.1944 | -2.0771 | 0.5945 |
| Salicylic | 5000 | 4825 | 0.454 | 0.2541 | -1.6494 | 0.7126 |
| Toluene | 3500 | 2744 | 0.2641 | 0.2343 | -2.2781 | 0.8181 |
| Uracil | 4000 | 1657 | 0.3161 | 0.2293 | -2.0884 | 0.7278 |
| **CMU Motion Capture Dataset** | | | | | | |
| Basketball | 74400 | 71674 | 0.8507 | 0.2811 | 2.0209 | 3.7188 |
| Walk | 71300 | 68452 | 0.835 | 0.2907 | 1.9866 | 3.7633 |
| Run | 17050 | 16110 | 0.7884 | 0.3216 | 1.0581 | 3.2393 |
| **AdK Dataset** | | | | | | |
| AdK | 818133 | 401936 | 0.1761 | 0.2493 | -3.0401 | 1.2173 |

The additional data in Table 8 further supports the non-stationary characteristics observed in the datasets. In particular, the MD17 dataset exhibits higher p-values in certain molecules (e.g., Benzene), indicating stronger non-stationary behavior. The CMU Motion dataset also demonstrates significant non-stationary characteristics, particularly in the basketball and walking sequences, where the mean ADF statistics are relatively high. The AdK dataset shows a relatively low mean p-value, indicating weaker stationarity.

The combination of these statistics and the stationarity assessment approach provides a robust framework for analyzing non-stationary characteristics in various datasets.

## B.2 Hyperparameter Settings

This section presents the hyperparameter configurations used for training on different datasets. Table 9 summarizes the settings for MD17, CMU Motion, and AdK Protein datasets. The same model architecture is used across all datasets, but specific training configurations, such as the number of epochs, learning rate, and dataset splits, are adjusted to suit the characteristics of each dataset.

Table 9 presents the hyperparameter settings for each dataset. The learning rate and weight decay values are chosen based on dataset characteristics, with AdK Protein requiring a lower learning rate due to its complexity. The CMU Motion dataset contains different train-validation-test splits for walk and run, which are provided separately in the table. The number of layers, hidden dimension, and weight decay remain consistent across all datasets.

## B.3 Compute Resources

All experiments reported in this paper were run on a dedicated high-performance server. The system is equipped with a single Intel® Xeon® Platinum 8358P CPU clocked at 2.60 GHz (32 cores, 64

Table 9: Hyperparameter configurations for different datasets. The dataset splits for CMU Motion are provided separately for *walk* and *run*.

| Hyperparameter | MD17 | CMU Motion | AdK Protein |
|---|---|---|---|
| Epochs | 500 | 500 | 150 |
| Learning Rate (lr) | $5 \times 10^{-3}$ | $5 \times 10^{-3}$ | $5 \times 10^{-5}$ |
| Weight Decay | $1 \times 10^{-12}$ | $1 \times 10^{-12}$ | $1 \times 10^{-12}$ |
| Number of Layers | 4 | 4 | 4 |
| $\Delta$ Frame | 5 | 1 | 5 |
| Hidden Dimension | 16 | 16 | 16 |
| Train/Val/Test Split | [2:4:4] | *walk:* [22:12:12] *run:* [5:4:2] | [6:2:2] |

threads), 251 GiB of DDR4 main memory, and four NVIDIA H20 GPUs (each with 96 GiB of VRAM). We used NVIDIA driver version 550.120 and CUDA 12.4. This configuration remained constant across all training and evaluation runs.

Table 10: Compute resources used for all experiments.

| Resource | Specification |
| --- | --- |
| CPU | Intel(R) Xeon(R) Platinum 8358P @ 2.60 GHz |
| | 1 socket, 32 cores/socket (64 threads) |
| Memory | 251 GiB DDR4 RAM |
| GPUs | $4 \times$ NVIDIA H20 |
| | 96 GiB HBM3 VRAM each, 4.0 TB/s bandwidth |
| Driver | NVIDIA driver 555.50 |
| CUDA | CUDA 12.4 |

## C  More Experimental Results

### C.1  Experiments on Differential Orders

We further conduct the experiments incorporating each order differences in Table 11, and higher order information may lead potential over-fitting in the deep model and result suboptimal performance. Hence, in ES-NGNN, we only incorporate 1st-order and 2nd-order differencing.

Table 11: The experiments on incorporating different orders while pooling.

| | Aspirin | Benzene | Ethanol | Malonaldehyde | Naphthalene | Salicylic | Toluene | Uracil | Average |
| --- | --- | --- | --- | --- | --- | --- | --- | --- | --- |
| None(ESTAG) | 0.677 | 0.086 | 0.422 | 0.632 | 0.328 | 0.629 | 0.369 | 0.366 | 0.439 |
| 1st | 0.467 | 0.726 | 0.419 | 0.634 | 0.368 | 0.45 | 0.33 | 0.381 | 0.39 |
| 2nd | 0.421 | 0.05 | 0.407 | 0.6 | 0.27 | 0.387 | 0.308 | 0.379 | **0.353** |
| 3rd | 0.501 | 0.058 | 0.408 | 0.583 | 0.273 | 0.341 | 0.3 | 0.4 | 0.358 |
| 4th | 0.509 | 0.067 | 0.419 | 0.607 | 0.302 | 0.352 | 0.278 | 0.425 | 0.37 |

### C.2  Ablations study that completely omitting Fourier Transform

Table 12: The ablation studies on removing PFT but keeping FT and completely omitting the FT.

| | Aspirin | Benzene | Ethanol | Malonaldehyde | Naphthalene | Salicylic | Toluene | Uracil | Average |
| --- | --- | --- | --- | --- | --- | --- | --- | --- | --- |
| NS-EGNN | 0.421 | 0.05 | 0.407 | 0.6 | 0.27 | 0.387 | 0.308 | 0.379 | **0.353** |
| NS-EGNN w/o PFT | 0.948 | 0.076 | 0.547 | 0.876 | 0.360 | 0.499 | 0.431 | 0.504 | 0.530 |
| NS-EGNN w/o FT | 0.564 | 0.062 | 0.436 | 0.619 | 0.385 | 0.425 | 0.358 | 0.467 | 0.480 |

We further conduct the ablation study that completely omitting the FT. The results are shown in Table 12. Surprisingly, we find totally remove FT even can outperform EGNN with FT, which indicates FT cannot accurately extract the intrinsic spectral information. The experiments demonstrates this inaccurate spectral feature also harms the convergence of the model.

### C.3  Additional non-stationary and equivariant normalization baselines

### C.4  Ablations study that completely omitting Fourier Transform

equivariant normalization [31]

### C.5  Experiments on Original Settings in ESTAG

We further present the results of the original setting of ESTAG in Table 14. NS-EGNN still outperforms ESTAG in 7 out of 8 cases under the settings in ESTAG, achieving 10.3% relative performance improvement in average.

Table 13: The ablation studies on removing PFT but keeping FT and completely omitting the FT.

| | Aspirin | Benzene | Ethanol | Malonaldehyde | Naphthalene | Salicylic | Toluene | Uracil | Average |
|---|---|---|---|---|---|---|---|---|---|
| NS-EGNN | 0.421 | 0.05 | 0.407 | 0.6 | 0.27 | 0.387 | 0.308 | 0.379 | **0.353** |
| EGNN w/normalization | 4.286 | 1.238 | 1.298 | 4.661 | 1.084 | 1.824 | 0.762 | 1.471 | 2.078 |
| Non-stationary Transformer | 773.835 | 351.856 | 526.389 | 853.422 | 1449.999 | 18.570 | 290.497 | 9.793 | 531.483 |

Table 14: The performance (MSE) of NS-EGNN in the setting of ESTAG.

| | **Aspirin** | **Benzene** | **Ethanol** | **Malonaldehyde** | **Naphthalene** | **Salicylic** | **Toluene** | **Uracil** | **Average** |
|---|---|---|---|---|---|---|---|---|---|
| ESTAG | 0.063 | 0.003 | 0.099 | 0.101 | 0.068 | **0.047** | 0.079 | 0.066 | 0.068 |
| NS-EGNN | **0.052** | 0.003 | **0.097** | **0.100** | **0.059** | 0.057 | **0.065** | **0.058** | **0.061** |

## C.6 Sensitivity of the Hyperparameters

Since the window length must be a factor of the number of past frames, and the hop length is typically selected as half of the window length to effectively capture dynamic frequencies, we explored various combinations of hop lengths for NS-EGNN, which applies PFT multiple times with different window lengths. The combinations are outlined as in Table 15.

Table 15: The ADE results of NS-EGNN with different set of hop length on MD17 dataset.

| **Hop** | **Aspirin** | **Benzene** | **Ethanol** | **Malonaldehyde** | **Naphthalene** | **Salicylic** | **Toluene** | **Uracil** | **Average** |
|---|---|---|---|---|---|---|---|---|---|
| [2, 5, 10] | 0.497 | 0.044 | 0.378 | 0.585 | 0.261 | 0.37 | 0.286 | 0.346 | 0.346 |
| [5, 10, 20] | 0.421 | 0.05 | 0.407 | 0.6 | 0.27 | 0.387 | 0.308 | 0.379 | 0.353 |
| [10, 20, 50] | 0.446 | 0.049 | 0.392 | 0.627 | 0.35 | 0.458 | 0.329 | 0.345 | 0.374 |

As observed in Table 15, these hyperparameters do not significantly impact the overall model performance.

## C.7 Complexity and Epoch Training Time

Theoretically, the attention mechanism requires $O(N^2)$ complexity, while the Fast Fourier Transform (FFT) only requires $O(N \log N)$ complexity. Experimentally, we measured the average per-epoch training time (in milliseconds) of NS-EGNN and the baselines on the MD17 dataset across seven molecules. The results in Table 16 indicate that NS-EGNN is the most efficient algorithm compared with the baselines.

## C.8 Sensitivity of Window Function

We further conduct the experiments on Blackman and Hann window functions on MD17 in Table 17. As observed in Table 17, NS-EGNN is insensitive to the type of window function. Additionally, the worst performance in the Table still outperforms all the baselines.

# D Limitation

On the ADK dataset, our method shows smaller improvements than on MD17 and CMU Motion. This happens because our approach is designed for non-stationary data and is less effective when the data are more stationary.

# E Broader Impacts

Our work contributes to more accurate and efficient dynamics simulations.

Broader impacts include:

- **Accelerating drug discovery:** Faster, more accurate simulations can reduce time and cost in identifying candidate compounds.

Table 16: Average epoch training time (ms) on MD17.

| Method | Aspirin | Benzene | Ethanol | Malonaldehyde | Naphthalene | Salicylic | Toluene | Uracil | Average |
|---|---|---|---|---|---|---|---|---|---|
| ESTAG | 40.3 | 21.1 | 12.9 | 17.5 | 32.7 | 32.6 | 23.8 | 21.8 | 25.34 |
| Eqmotion | 159.0 | 157.9 | 17.2 | 142.5 | 160.3 | 160.3 | 161.6 | 120.8 | 134.95 |
| AGL-STAN | 299.4 | 305.1 | 281.9 | 277.5 | 281.9 | 281.5 | 280.4 | 219.8 | 278.44 |
| NS-EGNN | 20.7 | 12.5 | 11.0 | 11.4 | 18.0 | 17.9 | 13.8 | 13.3 | 14.83 |

Table 17: The impact of different window functions on MD17 dataset.

| Window | Aspirin | Benzene | Ethanol | Malonaldehyde | Naphthalene | Salicylic | Toluene | Uracil | Average |
|---|---|---|---|---|---|---|---|---|---|
| Hamming | 0.409 | 0.038 | 0.389 | 0.584 | 0.32 | 0.338 | 3.291 | 0.331 | 0.337 |
| Blackman | 0.445 | 0.046 | 0.389 | 0.59 | 0.244 | 0.365 | 0.281 | 0.374 | 0.342 |
| Hann | 0.421 | 0.05 | 0.407 | 0.6 | 0.27 | 0.387 | 0.308 | 0.379 | 0.353 |

- **Environmental chemistry:** Improved modeling of reaction pathways may aid in designing greener catalysts and processes.

