# OpenReview forum: "Non-stationary Equivariant Graph Neural Networks for Physical Dynamics Simulation"
_NeurIPS.cc/2025/Conference — NeurIPS 2025 poster_

### Official Review · Reviewer_6j6b · 2025-06-23

**Clarity:** 2
**Significance:** 3
**Originality:** 2
**Rating:** 4
**Confidence:** 4

**Summary:**

This paper introduces NS-EGNN, a non-stationary equivariant graph neural network designed for modeling physical dynamics. NS-EGNN leverages Fourier Transforms to extract time-varying frequency features and applies first- and second-order differencing to mitigate linear and quadratic trends in the data. A pooling mechanism is employed to support future prediction. The proposed framework demonstrates consistent improvements over state-of-the-art baselines across various physical domains, including molecular, motion, and protein dynamics.

**Questions:**

- In Section 2.2, why not directly set $n=3$?
- What is difference between $s$, $\vec{s}$ and $S$, and what are their corresponding dimensions?
- In Equ (7), why does the overall time remain $T$ after applying the windowing function?
- Please check line 142 and 149, both sentences appear to be incomplete.

**Ethical Concerns:**

["NO or VERY MINOR ethics concerns only"]

**Final Justification:**

This paper presents a novel method and provides sufficient experimental results to demonstrate its effectiveness. Additionally, the authors have committed to addressing the typos and inconsistency in symbol usage. Based on these considerations, I am raising my score to Borderline Accept.

**Limitations:**

- The methodology section lacks clarity, making it difficult to follow. A general rephrasing is recommended. In particular, symbol usage should be carefully reviewed for consistency (see questions above). It is unclear how the PFT module is integrated with the NS pooling module, and further clarification is needed.
- An ablation study is necessary to validate the individual contributions of the PFT and NS pooling modules.
- The reported experimental results for ESTAG are weaker than those in the original paper. The authors should clarify this discrepancy.

**Quality:**

3

**Strengths And Weaknesses:**

- The proposed framework is conceptually novel and addresses an important challenge in modeling non-stationary physical dynamics.
- This paper provides some theorical guarantees for the equivariance and Invariance of the extracted features, that is appreciated.

---

> ### Author Rebuttal · Authors · 2025-07-31
>
> Thank you for your thorough and constructive feedback. We appreciate your time and insights, and will address all points below. Corresponding revisions will be incorporated in the manuscript.
>
> **Q1: In Section 2.2, why not directly set n=3?**
>
> We acknowledge that setting n=3 in Section 2.2 would be appropriate to maintain consistency with the E(3) group mentioned earlier. We will clarify this choice in the revised manuscript.
>
> **Q2: What is difference between $s$, $\vec{s}$ and 𝑆, and what are their corresponding dimensions?**
>
> We apologize for the notational confusion. For clarity:
>
> * $\vec{s} \in \mathbb{R}^{n\times T \times 3}$: Frequency information extracted by DFT.
> * $S \in \mathbb{R}^{n\times r \times 3}$: Frequency information extracted by PFT.
> * $s \in \mathbb{R}^{n\times r}$: Normalized version of $S$.
>
> We realize it may be more appropriate to change $S$ as $\vec{S}$ to avoid misunderstanding, and this will be reflected in our revised manuscript.
>
> **Q3: In Equ (7), why does the overall time remain 𝑇 after applying the windowing function?**
>
> In signal processing, instead of directly cutting off the signal, windowing functions (e.g., Hamming, Hann) attenuate signal discontinuities at segment boundaries *without altering the temporal length* \(T\). This preserves the time dimension while minimizing spectral leakage, consistent with standard signal processing practice. We will add a brief clarification to the manuscript.
>
> **Q4: Please check line 142 and 149, both sentences appear to be incomplete.**
>
> Thank you for catching these oversights:
> - Line 142: Corrected to "$\bar{\vec{X}}_{i}(t)$ is the average 142 position of all nodes in the graph."
> - Line 149: Revised to "However, directly cutting off parts of the trajectory can result in spectral leakage."
>
> **W1: The methodology section lacks clarity**
>
> - **Symbol usage should be carefully reviewed for consistency**
> We will make the correponding changes mentioned above.
>
> - **How the PFT module is integrated with the NS pooling module**
> We combine these modules by the flow of Equation 3-5. We first extract the invariant features from PFT, and using EGNN to update the coordinates $\hat{X}$, then NS-Pooling is applied to make the final predication.
>
>
> **W1: The methodology section lacks clarity**
> We will improve clarity by addressing both aspects raised:
>
> 1. **Symbol consistency**: We will implement all notational adjustments discussed previously (particularly in Q2), ensuring uniform symbol usage throughout.
> 2. **Module integration**: The PFT and NS-pooling modules connect through(Equation 3-5):
>    - PFT first extracts invariant features
>    - These features update coordinates ($\hat{X}$) via EGNN
>    - NS-pooling then generates final predictions
> We will add a clear pipeline description in Section 3.3 to explicitly articulate this workflow.
>
>
> **W2: An ablation study is necessary to validate the individual contributions of the PFT and NS pooling modules.**
>
> We presented ablation studies in:
>
> - **Table 6 (PFT)**: Replacing PFT with standard DFT increases relative MSE by 50.14%, confirming PFT’s necessity.
> - **Appendix C.1, Table 11 (NS-pooling)**: Substituting ESTAG’s pooling (equivariant but stationary) improves relative MSE by 24.36%, validating the importance of NS-pooling.
>
> Notably, both ablated variants still outperform all baselines, underscoring the framework’s robustness.
>
> **W3: The reported experimental results for ESTAG are weaker than those in the original paper. The authors should clarify this discrepancy.**
>
> The discrepancy of ESTAG stems from different experimental settings. Our main results use a **more challenging multi-step forecasting setting** than ESTAG’s original work. For direct comparison, we include results under identical settings in **Appendix C.2 (Table 12)**, where our implementation matches ESTAG’s reported performance. In this setting, NS-EGNN **still** outperforms ESTAG in 7 out of 8 cases, and has 10.3% relative performance improvement in average. We will emphasize this distinction in the revised version.
>
> We sincerely appreciate your valuable suggestions, which have strengthened our work. All changes will be implemented in the revised manuscript.

---

### Official Review · Reviewer_ZoES · 2025-06-30

**Clarity:** 3
**Significance:** 2
**Originality:** 3
**Rating:** 3
**Confidence:** 4

**Summary:**

This paper proposes NS-EGNN, a Non-Stationary Equivariant Graph Neural Network, to better simulate physical dynamics. The core idea is to extract time-varying frequency information from segmented trajectories using a Patch Fourier Transform (PFT), followed by a non-stationary pooling layer employing first and second-order differencing for trend mitigation. The method is evaluated on molecular, motion capture, and protein dynamics datasets.

**Questions:**

The questions are listed above.

**Ethical Concerns:**

["NO or VERY MINOR ethics concerns only"]

**Limitations:**

This paper does not explicitly discuss its limitations, such as the method's constraints, and focuses solely on analyzing the experimental results.

**Paper Formatting Concerns:**

No formatting concerns.

**Quality:**

2

**Strengths And Weaknesses:**

Strengths:
- The paper clearly motivates the need to handle non-stationarity in physical dynamics, identifying specific limitations in existing equivariant GNN models.
- The methodology is grounded in established time series and signal processing techniques (Fourier Transform, differencing) combined with equivariant learning.
- The paper is organized logically, and clearly defines notation and distinctions.

Weakness:
- **Limited Conceptual Novelty**: The components leveraged (windowed Fourier Transform, differencing for non-stationarity, equivariant GNNs) are all well-known techniques in their respective domains. The main contribution is their integration and careful adaptation.
- **Incomplete baselines**: lacks the comparison with existing Non-Stationary methods, such as PG-ODE and others. Additionally, there may be issues with the baseline implementation, as the AGL-STAN results appear to deviate significantly.

---

> ### Author Rebuttal · Authors · 2025-07-31
>
> Thank you for your feedback and suggestions. We appreciate the opportunity to address your concerns. Below are our point-by-point responses:
>
>
> **W1: Limited Conceptual Novelty**
>
> Our work makes two key conceptual contributions:
> 1. We identify and formally address the previously overlooked *non-stationarity* in physical dynamical systems;
> 2. Due to the non-stationarity, a naive FT is not sufficient to capture the comprehensive frequency information. We introduce windowed FT to capture the dynamic frequency.
> 3. We establish novel theoretical foundations (Section 3.1.3) for integrating windowed FT within equivariant frameworks, which is a non-trivial extension requiring rigorous proof.
>
> We will further clarify these contributions in our manuscript revision.
>
> **W2.1: Incomplete baselines**
>
> We appreciate this valuable suggestion. To the best of our knowledge, PG-ODE is not currently open-sourced (we would be happy to include it as a baseline if codes are available), which makes direct comparison challenging. Recognizing the importance of non-stationary baselines, we have included another well-known non-stationary method, Non-stationary Transformer [1], for comparison. The results demonstrate NS-EGNN's significant advantages over existing non-stationary approach:
>
> ||Aspirin|Benzene|Ethanol|Malonaldehyde|Naphthalene|Salicylic|Toluene|Uracil|Average|
> |-|-|-|-|-|-|-|-|-|-|
> |Non-stationary Transformer|773.835|351.856|526.389|853.422|1449.999|18.570|290.497|9.793|531.483|
> |NS-EGNN|0.421|0.050|0.407|0.600|0.270|0.387|0.308|0.379|0.353|
>
> *Note: All values are MSE (lower is better)*
>
> This performance gap occurs because standard normalization breaks physical symmetry priors, inducing severe overfitting. We will include this baseline and emphasize this analysis in our revision. Additionally, we would also be pleased to include additional baselines within this realm.
>
> **W2.2: Issue of AGL-STAN implemetation**
>
> We confirm our AGL-STAN implementation was directly adapted from the official ESTAG repository without any modification. As noted in the ESTAG paper, AGL-STAN underperforms equivariant methods due to its non-equivariant architecture. The significant deviation in AGL-STAN's results arises because it lacks equivariance considerations, making it prone to overfitting in physical dynamics scenarios. We will clarify this methodological distinction in the Experiment section.
>
> **Limitation Discussion**
>
> Thank you for highlighting this. We have discussed:
> (1) Broader limitations in Appendix D, and
> (2) NS-EGNN's specific constraints regarding complex graph relations (Section 3.2).
> In revision, we will expand these discussions and explicitly note that advanced graph backbones may be required for highly intricate systems. Furthermore, we maintain that these limitations do not diminish our core contribution of addressing non-stationarity in physical dynamic systems.
>
> We thank you again for your valuable insights. All suggested improvements will be reflected in the revised manuscript.
>
> [1]. Non-stationary Transformers: Exploring the Stationarity in Time Series Forecasting

---

> > ### Author Response · Authors · 2025-08-05
> > **Looking forward to your feedback**
> >
> > Dear Reviewer,
> >
> > As the discussion deadline approaches, we would greatly appreciate your feedback to deepen our discussion. If you still have any additional suggestions, please do not hesitate to let us know. We are eager to address any concerns more thoroughly.
> >
> > Thank you once again for dedicating your valuable time to reviewing our work.
> >
> > Best regards,
> >
> > The Authors

---

> > > ### Author Response · Authors · 2025-08-07
> > > **Looking forward to your feedback**
> > >
> > > Dear Reviewer ZoES,
> > >
> > > Thank you for your time and effort in reviewing our work. We are respectfully following up as the discussion period concludes in approximately 40 hours.
> > >
> > > We kindly ask you to review our rebuttal at your earliest convenience to see if they have addressed your concerns. If our responses resolve your concerns, we would appreciate it if you could consider updating your scores.
> > >
> > > Thank you again for dedicating your valuable time to reviewing our work.
> > >
> > > Best regards,
> > >
> > > The Authors

---

> > ### Comment · Reviewer_ZoES · 2025-08-07
> >
> > Thank authors for the further clarification and extensive efforts. Some of my concerns have been addressed, but the conceptual novelty is still limited. I will maintain the score.

---

### Official Review · Reviewer_2Kiq · 2025-07-01

**Clarity:** 1
**Significance:** 2
**Originality:** 3
**Rating:** 4
**Confidence:** 3

**Summary:**

This paper proposes Non-Stationary Equivariant Graph Neural Network (NS-EGNN) which uses patch Fourier transform to encode temporally varying features, equivariant GNN to encode spatial features, and temporal differences to predict non-stationary processes.  The method is evaluated on molecular dynamics, motion capture, and protein dynamics.

**Questions:**

- I'm not sure about the claim that "most existing model focus on single-step frame-to-frame forecasting." In my experience (in fluids or robotics ,e.g.), many time series prediction method encode many past frames and predict many future frames.  Maybe this is just in MD?
- The paper makes the claim that "normalization" is the typical method for dealing with non-stationarity and that this is incompatible with equivariance.  Can you explain further?  Many normalization schemes are using for equivariant architechures, so maybe it is just naive normalization which is problematic?
- Do the baselines encode multiple past time steps?  What about an ablation where the fourier transform is omitted completely and replaced with a encoder of enough past time steps to encode and track the non-stationarity?
- I am not clear how the NS-pooling operation works with the model.  I understand equation 10 as a mapping X^(L) to X^*.  From Eqn. 5, it seems this is being applied to the outputs of the EGNN layer.  Does this mean the EGNN outputs X^(L)  or does the EGNN output \DetlaX and \Delta^2 X.  If the former, it's not clear to me how this achieves the goal of making the NN prediction stationary.
- On line 177, is the FFT O(N logN) ?

**Ethical Concerns:**

["NO or VERY MINOR ethics concerns only"]

**Final Justification:**

I appreciate the author clarifications.  The rebuttal is quite good.  I particularly like the new ablations, clarifications on method, and explanations to help motivate the choices and novelty.  I am happy to raise my score.  (It's too bad you can't submit a revision -- it would be easier to evaluate clarity revisions.)

**Limitations:**

Yes.

**Paper Formatting Concerns:**

No concerns.

**Quality:**

2

**Strengths And Weaknesses:**

### Strengths
- The paper addresses an important problem.  Non-stationary processes are commonly encountered in many time series.  Additionally many physical dynamics problems have symmetries which can be used to inform and improve prediction.
- So far as I can determine the method is novel and appropriate to the problem.  Most other equivariant GNNs do not address non-stationarity and the experimental results back up the hypothesis that they will not perform that well out-of-the-box.
- The datasets and baselines are reasonable and the model outperform the baselines.
- The ablation shows the value of many of the design choices, but I would still like to see more ablations for example without the multi-scale and using no FT at all.

### Weaknesses
- The paper presentation is not great.  There are many spelling errors, missing words, and typos. The number is high enough that it is not just a matter of polish; the ability to understand the paper is compromised.
- In particular, the description of the method in Section 3 is not very clear.  The indices and variable names are not clear.  For example in Eqn. 2, the index t doesn't appear in the variables. In Eqn. 3-5, \vec{X} is redefined several times, making this hard to follow.  NS-Pooling is defined here, but not used in 3.1.2.  In 3.1.2 the notation \vec{X}^* is used but not elsewhere, and so on.
- While I found the method hard to understand, I also found it to involve several design choices for which I didn't understand the a priori motivation or see the empirical justification for.  For example, FT converts the signal to global frequencies which are by definition not temporally localized.  The PFT seems a bit ad hoc to me.  What is the performance of the method if we skip the FT entirely and just use encoded patches?  Could we use something temporally localized as well like wavelet transform?  I think the motivation to compute the norm in Eqn. 8 is to create an invariant feature to preserve equivariance in EGNN, but how much frequency information is preserved?  Could we use other norms directly on the original signal?  Is the Multi-scale PFT new or an existing idea? Is there theoretical support for multi-scale PFT? Does it correspond in some way to the scale-translation group?
- It would be good to give some intuitive grounding the p-value and ADF Mean for example by computing them for some synthetic stochastic processes which can be visualized.

---

> ### Author Rebuttal · Authors · 2025-07-31
>
> We sincerely thank the reviewer for the thorough evaluation and constructive feedback. We have carefully addressed each point below and will incorporate all changes in the final manuscript.
>
> **W1: The paper presentation is not great.**
>
> We apologize for any typographical errors in the current draft. We will conduct thorough proofreading to enhance readability and ensure all notations are consistent in the revised version.
>
> **W2: The description of the method in Section 3 is not very clear**
>
> We regret any confusion caused by the notation. To clarify:
>
> - Equation 2: $\vec{X}$ will be updated to $\vec{X}(t)$.
> - Equations 3-5 will be revised for consistency:
>
> $s = \text{PFT}(\{{\vec{\mathbf{X}}(t)}\}_{t=0}^{T})$
>
>
> $(h, s, \{\vec{\mathbf{X}}(t)^{(L)}\}\_{t=0}^{T}) = \text{EGNN}(h, s, \{\vec{\mathbf{X}}(t)\}\_{t=0}^{T})$
>
> $\vec{\mathbf{X}}^* = \text{NS-Pooling}(\{{\vec{\mathbf{X}}(t)}\}_{t=0}^{T})$
>
> Here, ${\mathbf{X}}(t)^{(L)}$ denotes the $L$-th EGNN layer output, and $\vec{\mathbf{X}}^*$ is the final pooled representation.
> - NS-Pooling remains conceptual as represented in Equation 10.
> - $\vec{X}^*$ is further utilized to calculate the MSE loss (Section 3.1, line 135).
>
> We are grateful for this feedback, which has strengthened our methodological exposition.
>
> **W3: The motivation of the design choices:**
>
> Since there are several sub-questions in this part. We carefully address each question individually.
>
> **W3.1 What is the performance if we skip the FT entirely and just use encoded patches?**
>
> We further conduct the ablation study that completely omitting the FT. Please refer to our response of Q3.2 below.
>
> **W3.2 Could we use something temporally localized as well like wavelet transform?**
>
> No, we cannot directly employ wavelet transform in our framework. While wavelet transforms are valuable for localized temporal analysis, they present challenges in equivariant settings:
> - The calculation of wavelet basis requires the operations of scaling and translation, disrupting the E(3) equivariance.
> - Subsampling in discrete wavelets breaks translation equivariance.
>
> Our PFT approach maintains strict E(3) equivariance required for physical dynamics. That said, we recognize wavelet transforms as a powerful alternative for non-stationary signal analysis. Future research directions could investigate modified wavelet constructions that maintain equivariance properties, and we will explicitly include this promising avenue in our future work discussion.
>
> **W3.3 How much frequency information is preserved?**
>
> Preservation depends critically on:
> 1) Hop length reduction to increase window overlap [1]
> 2) Diverse window lengths to capture multi-resolution spectra [2]
>
> Practically, high overlap combined with multi-scale sampling enables comprehensive frequency retention.
>
> **W3.4 Could we use other norms directly on the original signal?**
>
> Yes, we include an equivariant normalization method as a baseline in Q2.
>
> **W3.5 Is the Multi-scale PFT new or an existing idea?**
>
> Yes, multi-scale PFT is a new idea. While windowed Fourier analysis exists in signal processing field (e.g., wavelet transforms use a similar ideology), our contributions are:
> 1) First adaptation to E(3)-equivariant machine learning scenarios for non-stationary dynamics
> 2) Providing the method to integrate PFT with geometric GNNs while keeping equivariance of the framework
> 3) Flexible window-length mechanism surpassing wavelet constraints
>
>
> **W3.6 Is there theoretical support for multi-scale PFT?**
>
> Regarding the equivariance property of the multi-scale PFT, we provide a rigorous proof of Theorem 3.2. From the perspective from spectral retention, empirically, we observe that NS-EGNN with diverse window lengths and high overlap rates captures more comprehensive spectral features. However, while we can experimentally demonstrate multi-scale PFT's effectiveness, we currently lack a theoretical framework to quantitatively analyze spectral information retention.
>
> **W3.7 Does it correspond in some way to the scale-translation group?**
>
> No, PFT does not demonstrate scale equivariance but E(3) equivariance (including translation). As established in Theorem 3.2, PFT is E(3)-equivariant (translation/rotation). Scale equivariance doesn't hold: scaling trajectories proportionally scales $s$, breaking EGNN equivariance.
>
> From an application perspective, we argue that NS-EGNN should not inherently possess scale equivariance. This is because changes in scale (manifested as altered edge lengths) typically correspond to meaningful physical property changes in the system.
>
> **W4: Visualize the meaning of p-value and ADF Mean**
>
> To give a brief understanding of ADF test, we here provide two simplest examples to illustrate the calculation of ADF value and p-value. However, regretfully, since we are not able to upload image, we would include the illustration of example processes and detailed calculation in the Appendix.
>
> 1. **Process Generation**:
>    - Stationary: $X_t = 0.6X_{t-1} + e_t, e_t \in \mathcal{N}(0, 1)$
>    - Non-stationary: $X_t = X_{t-1} + e_t, e_t \in \mathcal{N}(0, 1)$
> 2. **ADF Test**:
>    - Regression: $\Delta X_t = \alpha + \beta X_{t-1} + \gamma \Delta X_{t-1} + e_t$
>    - Test statistic: $\text{ADF} = \hat{\beta}/\text{SE}(\hat{\beta})$
>      - Stationary: $\hat{\beta} = -0.4 \rightarrow \text{ADF} = -4.3$
>      - Non-stationary: $\hat{\beta} = -0.02 \rightarrow \text{ADF} = -1.1$
> 3. **p-value Interpretation**:
>    - Stationary: p = 0.001 (strong evidence against null)
>    - Non-stationary: p = 0.72 (weak evidence against null)
>
> **Q1: 'm not sure about the claim that "most existing model focus on single-step frame-to-frame forecasting." In my experience (in fluids or robotics ,e.g.), many time series prediction method encode many past frames and predict many future frames. Maybe this is just in MD?**
>
> Our statement primarily refers to the specific context of geometric graph neural networks for physical dynamics simulation, where many existing approaches (particularly those cited in our related works section) do indeed focus on single-step prediction.
>
> We will clarify the domain specificity in our revised claims.
>
> **Q2: Is normalization incompatible with equivariance?**
>
> Indeed, as demonstrated by recent work [3], naive normalization method is incompatible with equivariance. However, we also find a more advanced normalization method maintaining equivariance, and we have included it as our baseline [3] (without FT). The results are below:
>
> ||Aspirin|Benzene|Ethanol|Malonaldehyde|Naphthalene|Salicylic|Toluene|Uracil|Average|
> |-|-|-|-|-|-|-|-|-|-|
> |NS-EGNN|0.421|0.050|0.407|0.600|0.270|0.387|0.308|0.379|0.353|
> |EGNN w/ normalization|4.286|1.238|1.298|4.661|1.084|1.824|0.762|1.471|2.078|
>
>
> The results clearly demonstrate that normalization significantly degrades model performance (average error increases from 0.353 to 2.078). This performance deterioration stems from normalization's dual effect: while it helps stationarize the trajectories, it simultaneously removes critical temporal patterns that are essential for accurate predictions.
>
>
> **Q3.1: Do the baselines encode multiple past time steps**
>
> Yes, ESTAG employs an attention layer to encode the multiple past time steps. Other baselines are not designed for time-series in their original paper, and these baselines are modified by encoding the past time steps by linear layers.
>
> **Q3.2: An ablation where the FT is omitted completely**
>
> We further conduct the ablation study that completely omitting the FT. The results can be presented below:
>
> ||Aspirin|Benzene|Ethanol|Malonaldehyde|Naphthalene|Salicylic|Toluene|Uracil|Average|
> |-|-|-|-|-|-|-|-|-|-|
> |NS-EGNN|0.421|0.050|0.407|0.600|0.270|0.387|0.308|0.379|0.353|
> |NS-EGNN w/ FT|0.948|0.076|0.547|0.876|0.360|0.499|0.431|0.504|0.530|
> |NS-EGNN w/o FT|0.564|0.619|0.436|0.619|0.385|0.425|0.358|0.467|0.480|
>
> Surprisingly, we find totally remove FT even can outperform EGNN with FT, which indicates FT cannot accurately extract the intrinsic spectral information. The experiments demonstrates this inaccurate spectral feature also harms the convergence of the model.
>
> **Q4: How the NS-pooling operation works with the model to make the output stationary?**
>
> We apologize for any misundetstanding. We clarify:
> 1. The output of EGNN is $\{{\vec{\mathbf{X}}(t)}\}_{t=0}^{T}$.
> 2. NS-Pooling derives *stationarized* $\Delta \vec{\mathbf{X}}$ and $\Delta^2 \vec{\mathbf{X}}$ from the output of EGNN $\{{\vec{\mathbf{X}}(t)}\}_{t=0}^{T}$ (Equation 9).
> 3. This stationarization improves NN generalization by providing NN more stationary features.
>
> Please kindly note that our goal is stationarizing *inputs*, not outputs (predictions actually should be inherently non-stationary).
>
>
> **Q5: On line 177, is the FFT O(N logN)?**
>
> Correct. We will update this in the manuscript. Thanks for pointing this typo.
>
> **Q6: Ablation of multi-scale PFT**
>
> We employ reduce utilizing multiple windows to only utilize single window, the results can be found below:
>
> ||Aspirin|Benzene|Ethanol|Malonaldehyde|Naphthalene|Salicylic|Toluene|Uracil|Average|
> |-|-|-|-|-|-|-|-|-|-|
> |NS-EGNN (multi-scale)|0.421|0.050|0.407|0.600|0.270|0.387|0.308|0.379|0.353|
> |NS-EGNN (single-scale) PFT|0.454|0.058|0.415|0.598|0.301|0.410|0.309|0.392|0.366|
>
> Multi-scale PFT consistently outperforms single-scale, validating our design.
>
> All discussed modifications will be implemented in the revision. We thank the reviewer again for their insightful feedback that has significantly improved our work.
>
> [1]. Window and Overlap Processing Effects on Power Estimates from Spectra. 2000
>
> [2]. A Simple Adjustable Window Algorithm to Improve FFT Measurements. 2002
>
> [3]. Towards Geometric Normalization Techniques in SE(3) Equivariant Graph Neural Networks for Physical Dynamics Simulations. IJCAI 2024

---

> > ### Author Response · Authors · 2025-08-05
> > **Looking forward to your feedback**
> >
> > Dear Reviewer,
> >
> > As the discussion deadline approaches, we would greatly appreciate your feedback to deepen our discussion. If you still have any additional suggestions, please do not hesitate to let us know. We are eager to address any concerns more thoroughly.
> >
> > Thank you once again for dedicating your valuable time to reviewing our work.
> >
> > Best regards,
> >
> > The Authors

---

> > > ### Comment · Reviewer_2Kiq · 2025-08-06
> > > **Thank you for updates and answers.**
> > >
> > > I appreciate the author clarifications.  The rebuttal is quite good.  I particularly like the new ablations, clarifications on method, and explanations to help motivate the choices and novelty.  I am happy to raise my score.  (It's too bad you can't submit a revision -- it would be easier to evaluate clarity revisions.)

---

> > > > ### Author Response · Authors · 2025-08-06
> > > >
> > > > Dear Reviewer 2Kiq,
> > > >
> > > > Thank you so much for your positive feedback on our rebuttal and for your decision to raise your score! We truly appreciate it. We're very glad to hear that you found the clarifications helpful, and particularly value:
> > > >
> > > > - Your recognition of our **new ablation studies**
> > > > - Your positive note on our **method clarifications**
> > > > - Your appreciation of our **novelty explanations**
> > > >
> > > > We are also deeply grateful for your understanding regarding our inability to submit a paper revision at this stage. Our committed revision plan includes:
> > > >
> > > > 1. **Methodology clarification**: Refining notations in Equations 2-5 for better readability.(W2) Visually illustrating the relationship between PFT and NS-Pooling in Figure 2(Q4). Adding a novelty summary(W3.5)
> > > >
> > > > 2. **Extended ablation studies**: Evaluating "only FT" configuration (Q3.2), Testing single-scale PFT(Q6)
> > > >
> > > > 3. **Enhanced motivation**: Clarifying claimed domains(Q1) Adding normalization baseline comparisons (Q2), Discussing other temporal localization methods (W3.2). Explaining "why not only FT" (Q3.2),
> > > >
> > > > 4. **Additional visualizations**: Providing intuitive illustrations of the ADF test (W4).
> > > >
> > > > We believe the additional motivation and methodology explanations provided in the rebuttal can be seamlessly incorporated into the final version for enhanced clarity.
> > > >
> > > > Thank you again for your time, thoughtful review, and this encouraging update.
> > > >
> > > > Best regards,
> > > >
> > > > The Authors

---

### Official Review · Reviewer_eQFU · 2025-07-03

**Clarity:** 3
**Significance:** 2
**Originality:** 2
**Rating:** 4
**Confidence:** 3

**Summary:**

This paper introduces a Non-Stationary Equivariant Graph Neural Network (NS-EGNN) to improve the modeling of physical dynamics by addressing the challenge of non-stationarity while preserving symmetric inductive bias. Unlike existing equivariant graph neural networks that ignore the time-varying nature of physical systems, NS-EGNN incorporates Fourier Transform on trajectory segments to extract time-varying frequency features. It further reduces non-stationarity through first- and second-order differencing, followed by temporal pooling for prediction. By capturing dynamic frequency patterns and mitigating linear and quadratic trends in the data, NS-EGNN effectively models temporal dependencies. The method is evaluated on diverse physical dynamics tasks—including molecular motion, human motion, and protein dynamics—and demonstrates superior performance over existing state-of-the-art models.

**Questions:**

See Weaknesses.

**Ethical Concerns:**

["NO or VERY MINOR ethics concerns only"]

**Final Justification:**

This paper introduces Fourier features into equivariant dynamic transformations, which I believe is a rather scientific way of incorporating prior knowledge. During the rebuttal phase, the authors also addressed my concerns. I recommend a score of 4.

**Limitations:**

YES

**Quality:**

3

**Strengths And Weaknesses:**

**Strengths**
- 1.Frequency Feature with local windows intuitively helps capture key features while avoiding redundant information.
- 2. The paper is clearly written and has strong reproducibility.

**Weaknesses and Questions**
- 1.The paper mentions that PFT is introduced because "static frequency information is not sufficient." Why is the traditional DFT considered as "static frequency," while PFT is regarded as "dynamic frequency"? Is there any experimental evidence proving that PFT performs better than DFT?
- 2.What is the additional time consumption introduced by PFT?
- 3.I would like to know the meaning of Equation (10). My understanding is that previous methods focused on predicting the conformation of the next frame, while your method additionally predicts the (approximate) second derivative and the conformation of the final frame. I would like to know the rationale behind this approach and whether the second derivative is used during inference. Predicting the final equilibrium conformation has also been proposed in many previous papers, such as eSCN [1] in their OC20 IS2RS [2] experiments. Should these works be discussed?

[1] Reducing SO(3) Convolutions to SO(2) for Efficient Equivariant GNNs

[2] The Open Catalyst 2020 (OC20) Dataset and Community Challenges

---

> ### Author Rebuttal · Authors · 2025-07-31
>
> Thanks for your valuable feedback! Please kindly find the detailed responses below.
>
> **Q1.1: Why is the DFT considered as "static frequency," while PFT is regarded as "dynamic frequency"?**
>
> Thank you for raising this point. The distinction stems from their temporal resolution:
> - **DFT** computes the global frequency spectrum of an entire signal, implicitly assuming stationarity (hence "static").
> - **PFT** analyzes short overlapping segments, capturing time-localized frequency evolution (thus "dynamic").
> We will clarify this terminology in Section 3.1 to avoid ambiguity.
>
> **Q1.2: Is there any experimental evidence proving that PFT performs better than DFT?**
>
> Yes, in ablation study (Section 4.5), we conduct an experiment to replace PFT with DFT, and we found the average performance decreases, (Averaged prediction error from 0.353 to 0.530).
>
> **Q2: What is the additional time consumption introduced by PFT?**
>
> We compared training times for 5 epochs across variants. The experiments are conducted on the machine introduced in Appendix B.3. As shown below, PFT adds minimal computational overhead:
>
> | |Aspirin|Benzene|Ethanol|Malonaldehyde|Naphthalene|Salicylic| Toluene|Uracil|Average|
> |-|-|-|-|-|-|-|-|-|-|
> |w/PFT|20.7|12.5|11.0|11.4|18.0|17.9|13.8|13.3|14.8|
> |w/o PFT|16.7|11.3|10.8|11.1|14.6|15.0|12.4|12.1|13.0|
>
> *Table: Training time (ms) per 5 epochs.*
>
> **Q3.1: The meaning of Equation (10)**
>
> We apologize for any confusion. Equation (10) $\vec{\mathbf{X}}_{i}^{*} = [\Delta \vec{\mathbf{X}}_i, \Delta^2 \vec{\mathbf{X}}_i] \cdot  \mathbf{\gamma} + \vec{\mathbf{X}}_{i}^{(L)}(T-1)$ integrates two components:
> 1. **Stationarized features**: $\Delta \vec{\mathbf{X}}_i$ (first-order differences from Eq. 9) and $\Delta^2 \vec{\mathbf{X}}_i$ (second-order differences) are projected via a linear layer $\mathbf{\gamma}$.
> 2. **Temporal context**: $\vec{\mathbf{X}}_{i}^{(L)}(T-1)$ denotes the EGNN module's final-layer output at $t=T-1$.
>
> This approach reduces non-stationarity by predicting incremental changes rather than absolute states. For non-stationary data, these incremental values may be more predictable. For example, in molecular dynamics, the model can more effectively learn energy changes within the system than directly learn features from absolute energy values.
>
> We will refine the explanation in Section 3.3.
>
>
> **Q3.2: Predicting the final equilibrium conformation has also been proposed in many previous papers, such as eSCN [1] in their OC20 IS2RS [2] experiments. Should these works be discussed?**
>
> We appreciate this valuable suggestion. While eSCN advances equivariant representations, its primary focus on equilibrium conformation prediction makes direct adaptation to molecular dynamics simulations non-trivial.
>
> **Key Difference: Dynamics vs. Static Structures** - MD simulations require modeling molecular motion over time, whereas equilibrium predictions provide only static "snapshot" structures.
>
> Additionally, eSCN analyzes individual frame conformations rather than *trajectory-based* forecasting. Since our baseline comparisons already include similar high-representation models (ST-TFN, ST-SE(3)-Transformer), we focused on directly comparable approaches. We will expand the discussion of conformation prediction's relation to our task in Section 5.
>
> We sincerely appreciate your valuable suggestions. We will include above discussions into our revised manuscript to enhance clarity.

---

> > ### Comment · Reviewer_eQFU · 2025-08-05
> >
> > Thank you for the response; I will maintain my original score.

---

> > > ### Author Response · Authors · 2025-08-05
> > > **Thanks for your response**
> > >
> > > Dear Reviewer eQFU,
> > >
> > > Thank you for your support of maintaining your positive assessment. We have carefully considered your valuable feedback and have endeavored to address all the concerns raised in your review through the additional experiments and clarifications. We believe these changes have strengthened the paper.
> > >
> > > If you feel the revisions have adequately addressed your points, we would be deeply grateful if you might consider a modest upward adjustment to your score.
> > >
> > > Thank you again for your valuable feedback and support.
> > >
> > > Best regards,
> > >
> > > The Authors

---

### Note · Authors · 2025-08-11

Dear Area Chair and Reviewers,

We sincerely appreciate the insightful feedback from all reviewers. We thank all reviewers for recognizing the strengths in our work: the importance of formulated problem (`Reviewers 2Kiq, 6j6b`); strong motivation (`Reviewers 2Kiq, ZoES`); conceptual novelty of our method (`Reviewer 6j6b`); reasonable methodology design (`Reviewers eQFU, 2Kiq`); theoretical guarantees (`Reviewer 6j6b`).

During the rebuttal, we have diligently addressed concerns through:

1. **Expanded ablation studies**:
   - PFT removal analysis (minimal time impact)
   - Omission of Fourier Transform
   - Single-scale PFT experiments

2. **Additional baselines**:
   - Non-stationary Transformer, which excels at modeling non-stationary time series.
   - GeoNorm, an equivariant normalization method

3. **Enhanced motivation**:
   We further discuss other possible designs:
   - Wavelet transform (breaking the equivariance)
   - Normalization (lossing details in non-stationary dynamics)
   - Conformation prediction methods (only predicting a final static state)

4. **Methodology clarification**:
   - Reorganizing the methodology explanation in Equation 3-5;
   - Explaining how PFT works with NS-Pooling

While `Reviewer eQFU` and `Reviewer 2Kiq` confirmed satisfaction with our responses, we note two concerns remain from `Reviewer ZoES` and `Reviewer 6j6b`.

1. Limited conceptual novelty(`Reviewer ZoES`):

Conceptually, our work reveals that real-world physical dynamics are highly non-stationary. Existing methods, such as wavelet transform (which breaks equivariance) and normalization (which loses non-stationary details), cannot adequately address such scenarios. Technically, NS-EGNN is not merely a simple combination of existing modules. We adapt the windowed Fourier Transform from signal processing while providing rigorous theoretical proofs to ensure equivariance, which constitutes a non-trivial and novel contribution.

2. Too extensive explanation(`Reviewer 6j6b`):

We have outlined a detailed revision plan to address this concern. We believe the rebuttal stage offers authors an opportunity to clarify their work further. Following our clarifications, all reviewers now understand our method's motivation and detailed design. We are therefore confident the revised manuscript will provide clear and concise explanations.

We sincerely thank all reviewers for their time and valuable feedback on our paper.

Best Regards,

The Authors.

---

### Decision · Program_Chairs · 2025-09-17

**Decision:**

Accept (poster)

**Comment:**

This paper proposes NS-EGNN, combining Patch Fourier Transform (PFT) with non-stationary pooling to address non-stationarity in physical dynamics while preserving E(3) equivariance. The method applies windowed Fourier transforms and differencing operations to extract time-varying frequency features for improved temporal modeling. The authors identify that existing equivariant GNNs inadequately handle non-stationary dynamics in physical systems. Their solution integrates signal processing techniques with geometric deep learning, providing theoretical guarantees for equivariance preservation. Experiments across molecular, motion, and protein dynamics show consistent improvements over baselines.

Strengths: Important problem addressing a gap in equivariant neural networks. Well-motivated approach supported by statistical analysis (ADF tests). Sound technical integration of established methods with novel adaptations for equivariant settings. Comprehensive experimental validation with thorough ablation studies demonstrating individual component contributions.

Weaknesses: Limited conceptual novelty since core components (windowed FT, differencing, equivariant GNNs) are well-established techniques. Initial presentation lacked clarity with inconsistent notation and incomplete explanations of module integration. Baseline comparisons could be more comprehensive for non-stationary methods.

Authors effectively addressed technical concerns with extensive additional experiments including PFT vs DFT comparisons, computational overhead analysis, and complete FT removal studies. Reviewers eQFU and 2Kiq raised scores after satisfactory responses. Reviewer ZoES maintained novelty concerns despite acknowledging improvements. Reviewer 6j6b appreciated clarifications but emphasized need for better initial presentation quality.

The paper makes a solid technical contribution by identifying and addressing an important gap in equivariant modeling. While individual components are known, their integration requires careful theoretical work to maintain equivariance properties. The experimental validation is thorough, and positive rebuttal responses with two score increases validate the merit. Despite novelty and presentation limitations, the technical soundness, problem importance, and comprehensive evaluation support acceptance.